# Covariate-Powered Empirical Bayes Estimation

**Nikolaos Ignatiadis**
Statistics Department
Stanford University
ignat@stanford.edu

**Stefan Wager**
Graduate School of Business
Stanford University
swager@stanford.edu

## Abstract

We study methods for simultaneous analysis of many noisy experiments in the presence of rich covariate information. The goal of the analyst is to optimally estimate the true effect underlying each experiment. Both the noisy experimental results and the auxiliary covariates are useful for this purpose, but neither data source on its own captures all the information available to the analyst. In this paper, we propose a flexible plug-in empirical Bayes estimator that synthesizes both sources of information and may leverage any black-box predictive model. We show that our approach is within a constant factor of minimax for a simple data-generating model. Furthermore, we establish robust convergence guarantees for our method that hold under considerable generality, and exhibit promising empirical performance on both real and simulated data.

## 1 Introduction

It is nowadays common for a geneticist to simultaneously study the association of thousands of different genes with a disease [Efron et al., 2001, Lönnstedt and Speed, 2002, Love et al., 2014], for a technology firm to have records from thousands of randomized experiments [McMahan et al., 2013], or for a social scientist to examine data from hundreds of different regions at once [Abadie and Kasy, 2018]. In all of these settings, we are fundamentally interested in learning something about each sample (i.e., gene, experimental intervention, etc.) on its own; however, the abundance of data on other samples can give us useful context with which to interpret our measurements about each individual sample [Efron, 2010, Robbins, 1964]. In this paper, we propose a method for simultaneous analysis of many noisy experiments, and show that it is able to exploit rich covariate information for improved power by leveraging existing machine learning tools geared towards a basic prediction task.

As a motivation for our statistical setting, suppose we have access to a dataset of movie reviews where each movie $i = 1, ..., n$ has an average rating $Z_i$ over a limited number of viewers; we also have access to a number of covariates $X_i$ about the movie (e.g., genre, length, cast, etc.). The task is to estimate the "true" rating $\mu_i$ of the movie, i.e., the average rating had the movie been reviewed by a large number of reviewers similar to the ones who already reviewed it. A first simple approach to estimating $\mu_i$ is to use its observed average rating as a point estimate, i.e., to set $\hat{\mu}_i = Z_i$. This approach is clearly valid for movies where we have enough data for sampling noise to dissipate, e.g., with over 50,000 reviews in the MovieLens 20M data [Harper and Konstan, 2016], we expect the 4.2/5 rating of Pulp Fiction to be quite stable. Conversely, for movies with fewer reviews, this strategy may be unstable: the rating 1.6/5 of Urban Justice is based on less than 20 reviews, and appears liable to change as we collect more data. A second alternative would be to just rely on covariates: We could learn to predict average ratings from covariates, $m(x) = \mathbb{E}\left[Z_i \,\middle|\, X_i = x\right]$, and then set $\hat{\mu}_i = \hat{m}(X_i)$. This may be more appropriate than using the observed mean rating for movies with very few reviews, but is limited in its accuracy if the covariates aren't expressive enough to perfectly capture $\mu_i$.

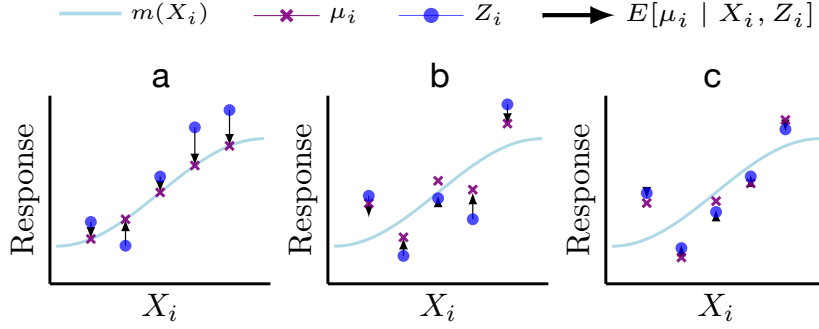

Figure 1: **Optimal empirical Bayes shrinkage.** All three plots show $\mu_i$ and $Z_i$ drawn from (1) for various values of $A/\sigma^2$, with the covariate values $X_i$ fixed and the regression curve $m(\cdot)$ shown in blue. The arrows depict how the oracle Bayes denoiser from (2) moves the point estimate $\hat{\mu}_i$ away from the raw observation $Z_i$ and towards $m(X_i)$. **a)** When $A/\sigma^2 = 0$, the oracle estimator shrinks $Z_i$ all the way back to $m(X_i)$. **b)** For $A/\sigma^2 = 1$, optimal shrinkage uses $(Z_i + m(X_i))/2$ to estimate $\mu_i$. **c)** When $A/\sigma^2$ is very large, it is preferable to discard $m(X_i)$ and just use the information in $Z_i$.

We develop an approach that reconciles (and optimally interpolates between) the two estimation strategies discussed above. The starting point for our discussion is the following generative model,

$$X_i \sim \mathbb{P}^X, \quad \mu_i \mid X_i \sim \mathcal{N}\left(m(X_i),\ A\right), \quad Z_i \mid \mu_i \sim \mathcal{N}\left(\mu_i,\ \sigma^2\right), \tag{1}$$

according to which the true rating $\mu_i$ of each movie is partially explained by its covariates $X_i$, but also has an idiosyncratic and unpredictable component with a Gaussian distribution $\mathcal{N}\left(0,\ A\right)$. Recall that we observe $X_i$ and $Z_i$ for each $i = 1,\ ...,\ n$, and want to estimate the vector of $\mu_i$. Given this setting, if we knew both the idiosyncratic noise level $A$ and $m(x)$, the conditional mean of $\mu_i$ given $X_i = x$, then the mean-square-error-optimal estimate of $\mu_i$ could directly be read off of Bayes' rule, $\hat{\mu}_i^* = t_{m,A}^*(X_i,\ Z_i)$, with

$$t_{m,A}^*(x, z) := \mathbb{E}_{m,A}\left[\mu_i \mid X_i = x,\ Z_i = z\right] = \frac{A}{\sigma^2 + A}z + \frac{\sigma^2}{\sigma^2 + A}m(x). \tag{2}$$

As shown in Figure 1, the behavior of this shrinker depends largely on the ratio $A/\sigma^2$: As this ratio gets large, the Bayes rule gets close to just setting $\hat{\mu}_i = Z_i$, whereas when the ratio is small, it shrinks everything to predictions made using covariates.

Now in practice, $m(\cdot)$ and $A$ are unlikely to be known a-priori and, furthermore, we may not believe that the hierarchical structure (1) is a perfect description of the underlying data-generating process. The main contribution of this paper is an estimation strategy that addresses these challenges. First, we derive the minimax risk for estimating $\mu_i$ in model (1) in a setting where $m(\cdot)$ is unknown but we are willing to make various regularity assumptions (e.g., that $m(\cdot)$ is Lipschitz). Second, we show that a feasible plug-in version of (2) with estimated $\hat{m}(\cdot)$ and $\hat{A}$ attains this lower bound up to constants that do not depend on $\sigma^2$ or $A$.

Finally, we consider robustness of our approach to misspecification of the model (1), and establish an extension to the classic result of James and Stein [1961], whereby without any assumptions on the distribution of $\mu_i$ conditionally on $X_i$, we can show that our approach still improves over both simple baselines $\hat{\mu}_i = Z_i$ and $\hat{\mu}_i = \hat{m}(X_i)$ in considerable generality (see Section 4 for precise statements). We also consider behavior of our estimator in situations where the distribution of $Z_i$ conditionally on $\mu_i, X_i$ may not be Gaussian, and the conditional variance $\sigma_i^2$ of $Z_i$ given $\mu_i, X_i$ may be different for different samples.

Our approach builds on a long tradition of empirical Bayes estimation that seeks to establish frequentist guarantees for plug-in Bayesian estimators and related procedures in data-rich environments [Efron, 2010, Robbins, 1964]. Empirical Bayes estimation in the setting without covariates $X_i$ is by now well understood [Brown and Greenshtein, 2009, Efron, 2011, Efron and Morris, 1973, Ignatiadis et al., 2019, Ignatiadis and Wager, 2019, James and Stein, 1961, Jiang and Zhang, 2009, Johnstone and Silverman, 2004, Muralidharan, 2010, Stephens, 2016, Weinstein et al., 2018].

In contrast, empirical Bayes analysis with covariates has been less comprehensively explored, and existing formal results are confined to special cases. Fay and Herriot [1979] introduced a model of the form (1) with a linear specification, $m(x) = x^\top \beta$, motivated by the problem of "small area estimation" that arises when studying small groups of people based on census data. Further properties of empirical Bayes estimators in the linear specification (including robustness to misspecification) were established by Green and Strawderman [1991] in the case where $X_i \in \mathbb{R}$ and $m(x) = x$, and by Cohen et al. [2013], Tan [2016] and Kou and Yang [2017] when $m(x) = x^\top \beta$. There has also been some work on empirical Bayes estimation with nonparametric specifications for $m$, e.g., Mukhopadhyay and Maiti [2004] and Opsomer et al. [2008]. In a genetics application, Stephan et al. [2015] parametrized $m(x)$ as a random forest. Banerjee et al. [2018] utilize univariate side information to estimate sequences of $\mu_i$ that consist mostly of zeros. We also note recent work by Coey and Cunningham [2019] who considered experiment splitting as an alternative to empirical Bayes estimation. Our paper adds to this body of knowledge by providing the first characterization of minimax-optimal error in the general model (1), by proposing a flexible estimator that attains this bound up to constants, and by studying robustness of non-parametric empirical Bayes methods to model misspecification.

## 2 Minimax rates for empirical Bayes estimation with covariates

We first develop minimax optimality theory for model (1), when $m$ is known to lie in a class $\mathcal{C}$ of functions. To this end, we formalize the notion of regret in empirical Bayes estimation, following Robbins [1964]. Concretely, as before, we assume that we have access to $n$ i.i.d. copies $(X_i, Z_i)$ from model (1); $\mu_i$ is not observed. Our task at hand then is to construct a denoiser $\hat{t}_n : \mathcal{X} \times \mathbb{R} \to \mathbb{R}$ that we will use to estimate $\mu_{n+1}$ by $\hat{t}_n(X_{n+1}, Z_{n+1})$ for a future sample $(X_{n+1}, Z_{n+1})$. We benchmark this estimator against the unknown Bayes estimator $t^*_{m,A}(X_{n+1}, Z_{n+1})$ from (2) in terms of its regret (excess risk) $L(\hat{t}_n; m, A)$, where:

$$L(t; m, A) := \mathbb{E}_{m,A} \left[ (t(X_{n+1}, Z_{n+1}) - \mu_{n+1})^2 \right] - \mathbb{E}_{m,A} \left[ \left( t^*_{m,A}(X_{n+1}, Z_{n+1}) - \mu_{n+1} \right)^2 \right] \quad (3)$$

We characterize the difficulty of this task by exhibiting the minimax rates for the empirical Bayes excess risk incurred by not knowing $m \in \mathcal{C}$ (but knowing $A$), where $\mathcal{C}$ is a pre-specified class of functions:[1]

$$\mathfrak{M}^{\mathrm{EB}}_n \left( \mathcal{C}; A, \sigma^2 \right) := \inf_{\hat{t}_n} \sup_{m \in \mathcal{C}} \left\{ \mathbb{E}_{m,A} \left[ L(\hat{t}_n; m, A) \right] \right\} \quad (4)$$

Our key result, informally stated, is that the minimax excess risk $\mathfrak{M}^{\mathrm{EB}}_n$ can be characterized in terms of the minimax risk for estimating $m(\cdot)$ with respect to $L^2(\mathbb{P}^X)$ in the regression problem in which we observe $(X_i, Z_i)_{1 \leq i \leq n}$ with $Z_i \mid X_i \sim \mathcal{N}(m(X_i), A + \sigma^2)$, i.e.,

$$\mathfrak{M}^{\mathrm{Reg}}_n \left( \mathcal{C}; A + \sigma^2 \right) := \inf_{\hat{m}_n} \sup_{m \in \mathcal{C}} \mathbb{E}_{m,A} \left[ \int \left( \hat{m}_n(x) - m(x) \right)^2 d\mathbb{P}^X(x) \right], \quad (5)$$

such that, for many commonly used function classes $\mathcal{C}$, we have [2]

$$\mathfrak{M}^{\mathrm{EB}}_n \left( \mathcal{C}; A, \sigma^2 \right) \asymp \frac{\sigma^4}{(\sigma^2 + A)^2} \mathfrak{M}^{\mathrm{Reg}}_n \left( \mathcal{C}; A + \sigma^2 \right). \quad (6)$$

In other words, when $A/\sigma^2$ is very large, we find that it is easy to match the performance of Bayes rule (2), since it collapses to $Z_i$. On the other hand, when $A/\sigma^2$ is small, matching the Bayes rule requires estimating $m(\cdot)$ well, and (6) precisely describes how the difficulty of estimating $m(\cdot)$ affects our problem of interest.

Previous work on minimax rates for the excess risk (3) has been sparse; some exceptions include Benhaddou and Pensky [2013], Li et al. [2005] and Penskaya [1995], who develop minimax bounds on (3) when $\mu \sim G$, $Z \mid \mu \sim \mathcal{N}(0, \sigma^2)$, i.e., in the setting without covariates but with potentially more general priors. Beyond the modulation through covariates, a crucial difference of our approach is that we pay attention to the behavior in terms of $A$ and $\sigma$, instead of absorbing them into constants.

**Lower bound** Here we provide a lemma for deriving lower bounds for worst case expected excess risk (4) through reduction to hypothesis testing. The result is applicable to any class $\mathcal{C}$ for which we can prove a lower bound on the minimax regression error using Le Cam's two point method or Fano's method [Duchi, 2019, Györfi et al., 2006, Ibragimov and Hasminskii, 1981, Tsybakov, 2008]; we will provide concrete examples below.

**Lemma 1.** *For each $n$, let $\mathcal{V}_n$ be a finite set and $\mathcal{C}_n = \{m_{n,v} \mid v \in \mathcal{V}_n\} \subset \mathcal{C}$ be a collection of functions indexed by $\mathcal{V}_n$ such that for a sequence $\delta_n > 0$:*

$$\int \left(m_{n,v}(x) - m_{n,v'}(x)\right)^2 d\mathbb{P}^X(x) \geq \delta_n^2 \text{ for all } v \neq v' \in \mathcal{V}_n, \text{ for all } n$$

*If furthermore, $\sup_{v,v' \in \mathcal{V}_n} \sup_x \left(m_{n,v}(x) - m_{n,v'}(x)\right)^2 \to 0$ as $n \to \infty$, then:*

$$\mathfrak{M}_n^{EB}\left(\mathcal{C}; A, \sigma^2\right) \gtrsim \frac{\sigma^4}{\left(\sigma^2 + A\right)^2} \cdot \delta_n^2 \cdot \inf_{\hat{V}_n} \mathbb{P}[\hat{V}_n \neq V_n]$$

*Here, $\inf_{\hat{V}_n} \mathbb{P}[\hat{V}_n \neq V_n]$ is to be interpreted as follows: $V_n$ is drawn uniformly from $\mathcal{V}_n$ and conditionally on $V_n = v$, we draw the pairs $(X_i, Z_i)_{1 \leq i \leq n}$ from model (1) with regression function $m_{n,v}(\cdot)$. The infimum is taken over all estimators $\hat{V}_n$ that are measurable with respect to $(X_i, Z_i)_{1 \leq i \leq n}$.*

The Lemma may be interpreted as follows: If information theoretically we cannot determine which $m_{n,v} \in \mathcal{C}_n$ generated $(X_i, Z_i)_{1 \leq i \leq n}$, yet the $m_{n,v}$ are well separated in $L^2(\mathbb{P}^X)$ norm, then the minimax empirical Bayes regret (4) must be large. Proving lower bounds involves contructing $\mathcal{C}_n$.

**Upper bound** Previously, we described the relationship of model (1) to nonparametric regression. However, there is a further connection: Under (1), it also holds that $Z_i \mid X_i \sim \mathcal{N}\left(m(X_i), \sigma^2 + A\right)$. Thus $m(\cdot)$ may estimated from the data by directly running a regression $Z_i \sim X_i$. Then, for known $A$, the natural impetus to approximate (2) in a data-driven way is to use a plug-in estimator. Concretely, given a $\hat{m}_n$ that achieves the minimax risk (5), we just plug that into the Bayes rule (2):

$$\hat{t}_n(x,z) := t^*_{\hat{m}_n, A}(x,z) = \frac{A}{\sigma^2 + A} z + \frac{\sigma^2}{\sigma^2 + A} \hat{m}_n(x) \tag{7}$$

This plug-in estimator, establishes the following upper bound on (4):

**Theorem 2.** *Under model (1), it holds that:*

$$\mathfrak{M}_n^{EB}\left(\mathcal{C}; A, \sigma^2\right) \leq \frac{\sigma^4}{\left(\sigma^2 + A\right)^2} \mathfrak{M}_n^{Reg}\left(\mathcal{C}; A + \sigma^2\right)$$

In deriving the lower bound Lemma (1), the estimators considered may use the unknown $A$. For this reason, for the upper bound we also benchmark against estimators that know $A$; however in Section 3 we demonstrate that in fact knowledge of $A$ is not required to attain optimal rates. Next we provide two concrete examples of classes, where the lower and upper bounds match up to constants.

**The linear class (Fay-Herriot shrinkage)** As a first, simple example, we consider the model of Fay and Herriot [1979], in which: $\mathcal{X} = \mathbb{R}^d$, and $\mathcal{C} = \text{Lin}\left(\mathbb{R}^d\right) = \left\{m \mid m(x) = x^\top \beta, \ \beta \in \mathbb{R}^d\right\}$.

**Theorem 3.** *Assume the $X_i$ are $\overset{iid}{\sim} \mathcal{N}(0, \Sigma)$ for an unknown covariance matrix $\Sigma \succ 0, \Sigma \in \mathbb{R}^{d \times d}$. Then there exists a constant $C_{Lin}$ (which does not depend on the problem parameters) such that:*

$$\lim_{n \to \infty} \left| \log\left(\mathfrak{M}_n^{EB}\left(Lin\left(\mathbb{R}^d\right); A, \sigma^2\right) \Big/ \frac{\sigma^4}{\left(\sigma^2 + A\right)^2} \cdot \frac{(\sigma^2 + A)d}{n}\right) \right| \leq C_{Lin}$$

**The Lipschitz class** Next we let $\mathcal{X} = [0,1]^d$ and for $L > 0$ we consider the Lipschitz class:

$$\mathcal{C} = \text{Lip}([0,1]^d, L) := \left\{m : [0,1]^d \to \mathbb{R} \mid |m(x) - m(x')| \leq L \|x - x'\|_2 \ \forall \, x, x' \in [0,1]^d\right\}.$$

**Theorem 4.** *Assume the $X_i$ are $\overset{iid}{\sim} F^X$, where $F^X$ is a measure on $[0,1]^d$ with Lebesgue density $f^X$ that satisfies $\eta \leq f^X(u) \leq 1/\eta$ for all $u \in [0,1]^d$ for some $\eta > 0$. Then there exists a constant $C_{Lip}(d, \eta)$ which depends only on $d, \eta$ such that:*

$$\lim_{n \to \infty} \left| \log\left(\mathfrak{M}_n^{EB}\left(Lip([0,1]^d, L); A, \sigma^2\right) \Big/ \frac{\sigma^4}{\left(\sigma^2 + A\right)^2} \cdot \left(\frac{L^d\left(\sigma^2 + A\right)}{n}\right)^{\frac{2}{2+d}}\right) \right| \leq C_{Lip}(d, \eta)$$

# 3 Feasible estimation via split-sample empirical Bayes

The minimax estimator in (7) that implements (2) in a data-driven way is not feasible, because $A$ is unknown in practice. In principle, $A + \sigma^2$ (with $\sigma^2$ known) is just $\mathrm{Var}\left[Z_i \mid X_i\right]$, hence deriving a plug-in estimator for $A$ just takes us to the realm of variance estimation in regression problems. But variance estimation for the general setting we consider here is a notoriously difficult problem, with only partial solutions available for very specific settings [e.g., Janson et al., 2017, Reid et al., 2016]. Furthermore, even for 1-dimensional smooth nonparametric regression the minimax rates for variance estimation may be slower than parametric [Brown and Levine, 2007, Shen et al., 2019].

Fortunately, it turns out that we do not need to accurately estimate $A$ in (1) in order for our approach to perform well. Rather, as shown below, if we naively read off an estimate of $A$ derived via sample splitting as in (8), we still obtain strong guarantees. Concretely, we study the following algorithm:

1. Form a partition of $\{1, \dots, n\}$ into two folds $I_1$ and $I_2$.
2. Use observations in $I_1$, to estimate the regression $m(x) = \mathbb{E}\left[Z_i \mid X_i = x\right]$ by $\hat{m}_{I_1}(\cdot)$.
3. Use observations in $I_2$, to estimate $A$, through the formula:

$$\hat{A}_{I_2} = \left( \frac{1}{|I_2|} \sum_{i \in I_2} \left( \hat{m}_{I_1}(X_i) - Z_i \right)^2 - \sigma^2 \right)_+ \tag{8}$$

4. The estimated denoiser is then $\hat{t}_n^{\mathrm{EBCF}}(\cdot, \cdot) = t^*_{\hat{m}_{I_1}, \hat{A}_{I_2}}(\cdot, \cdot)$.

We prove the following guarantee for this estimator. In particular, the following implies that if the minimax rate for regression (5) is slower than the parametric rate $1/n$ and if $|I_1|/n$ converges to a non-trivial limit, then our algorithm attains the minimax rate even when $A$ is unknown.

**Theorem 5.** *Consider a split of the data into two folds $I_1, I_2$, where $n_1 = |I_1|$, $n_2 = |I_2|$. Furthermore assume that $\hat{m}_{I_1}$ satisfies $\mathbb{E}_{m,A}[\hat{m}_{I_1}(X)^4 \mid \hat{m}_{I_1}] \le M$ almost surely for some $M < \infty$, where $X$ is a fresh draw from $\mathbb{P}^X$. Then the estimator $\hat{t}_n^{EBCF}$ satisfies the following guarantee:*

$$\mathbb{E}_{m,A}\left[ L\left( \hat{t}_n^{EBCF};\ m, A \right) \right] \le \mathbb{E}_{m,A}\left[ L\left( t^*_{\hat{m}_{n_1}, A};\ m, A \right) \right] + \frac{1}{n_2} O\left( 1 \right)$$

We emphasize that this result does not depend on $\widehat{A}$ from (8) being a particularly accurate estimate of $A$. Rather, what's driving our result is the following fact: If (1) holds, but we use (2) with $\tilde{m}(\cdot) \ne m(\cdot)$, then the choice of $\tilde{A}$ that minimizes the Bayes risk among all estimators of the form $t^*_{\tilde{m}, \tilde{A}}(\cdot, \cdot)$, $\tilde{A} \ge 0$ is not $A$, but rather (cf. derivation in Proposition 15 of the Appendix)

$$A_{\tilde{m}} := \mathbb{E}_{m,A}\left[ (\tilde{m}(X_{n+1}) - Z_{n+1})^2 \right] - \sigma^2 = A + \mathbb{E}_{m,A}\left[ (\tilde{m}(X_{n+1}) - m(X_{n+1}))^2 \right].$$

In other words, we're better off inflating the prior variance to account for the additional estimation error of $\tilde{m}(\cdot)$; and this inflated prior variance is exactly what's captured in (8).

# 4 Robustness to misspecification

So far, our results and estimator apply to Robbins' model [Robbins, 1964] in which (1) holds and we are interested in a estimating a future $\mu_{n+1}$. However, it is also of considerable interest to understand the behavior of empirical Bayes estimation when the specification (1) doesn't hold. In this section, we consider properties of our estimator under the weaker assumption that we only have a generic data-generating distribution for $(X_i, \mu_i, Z_i)$ of the form

$$(X_i, \mu_i) \sim \mathbb{P}^{(X_i, \mu_i)}, \quad \mathbb{E}\left[Z_i \mid \mu_i, X_i\right] = \mu_i, \quad \mathrm{Var}\left[Z_i \mid \mu_i, X_i\right] = \sigma^2, \tag{9}$$

and we seek to estimate the unknown $\mu_1, \dots, \mu_n$ underlying the experiments we have data for. The distributions indexed by $i$ are assumed to be independent, but need not be identical. This setting is sometimes referred to as the compound estimation problem [Brown and Greenshtein, 2009].

We proceed with a cross-fold estimator, which we call EBCF (empirical Bayes with cross-fitting), as follows: We split the data as above, but now also consider flipping the roles of $I_1$ and $I_2$ such that we can make predictions $\hat{\mu}_i$ for all $i = 1, \dots, n$ as

$$\hat{\mu}_i^{\mathrm{EBCF}} = t^*_{\hat{m}_{I_1}, \hat{A}_{I_2}}(X_i, Z_i) \text{ for } i \in I_2 \quad \& \quad \hat{\mu}_i^{\mathrm{EBCF}} = t^*_{\hat{m}_{I_2}, \hat{A}_{I_1}}(X_i, Z_i) \text{ for } i \in I_1.$$

This is a 2-fold cross-fitting scheme, which has been fruitful in causal inference [Chernozhukov et al., 2017, Nie and Wager, 2018, Schick, 1986] and multiple testing [Ignatiadis et al., 2016, Ignatiadis and Huber, 2018]. We also note that extensions to $k$-fold cross-fitting are immediate.

**SURE for empirical Bayes** The key property of our estimator that enables our approach to be robust outside of the strict model (1) is as follows. Let SURE$(\cdot)$ denote Stein's Unbiased Risk Estimate, a flexible risk estimator that is motivated by the study of estimators for $\mu_i$ in the Gaussian model $Z_i \sim \mathcal{N}(\mu_i, \sigma^2)$ [Stein, 1981]. Then, although our estimator was not originally motivated by SURE, one can algebraically verify that our estimator with a plug-in choice of $\hat{A}$ in fact minimizes SURE among all comparable shrinkage estimators (the same holds true with $I_1, I_2$ flipped):

$$\hat{A}_{I_2} = \left( \frac{1}{|I_2|} \sum_{i \in I_2} (\hat{m}_{I_1}(X_i) - Z_i)^2 - \sigma^2 \right)_+ \iff \hat{A}_{I_2} = \operatorname*{argmin}_{A \geq 0} \{ \text{SURE}_{I_2}(A) \},$$

$$\text{where } \text{SURE}_{I_2}(A) := \frac{1}{|I_2|} \sum_{i \in I_2} \left( \sigma^2 + \frac{\sigma^4}{(A+\sigma^2)^2}(Z_i - \hat{m}_{I_1}(X_i))^2 - 2\frac{\sigma^4}{A+\sigma^2} \right). \tag{10}$$

Furthermore, SURE has the following remarkable property in our setting: For any data-generating process as in (9) and any $A \geq 0$ [see also Jiang et al., 2011, Kou and Yang, 2017, Xie et al., 2012],

$$\mathbb{E}\left[ \text{SURE}_{I_2}(A) \mid X_{1:n}, \mu_{1:n} \right] = \frac{1}{|I_2|} \sum_{i \in I_2} \mathbb{E}\left[ \left( \mu_i - t^*_{\hat{m}_{I_1}, A}(X_i, Z_i) \right)^2 \mid X_{1:n}, \mu_{1:n} \right], \tag{11}$$

even when the distribution of $Z_i$ conditionally on $\mu_i$ and $X_i$ is not Gaussian. Putting (10) and (11) together, we find that we can argue using SURE that our estimator minimizes an unbiased risk estimate for the generic specification (9), despite the fact that our procedure was not directly motivated by SURE and SURE itself was only designed for Gaussian estimation.

**Gaussian data with equal variance and James-Stein property** To derive a first consequence of the above, let us first focus on a special case of (9), where $Z_i \mid (\mu_i, X_i) \sim \mathcal{N}(\mu_i, \sigma^2)$. Then the EBCF estimate satisfies the James-Stein property of strictly dominating the direct estimator $Z_i$ [James and Stein, 1961][3]. In other words, even if one has covariates $X_i$, which are uninformative, or one uses a really poor method for prediction, one still does no worse than just using $\hat{\mu}_i := Z_i$.

**Theorem 6** (James-Stein property). *Under the assumptions above and if $|I_1|, |I_2| \geq 5$, the proposed estimator $\hat{\mu}_i$ uniformly dominates the (conditional) maximum likelihood estimator $Z_i$, in other words for all $\mu_1, \ldots, \mu_n$ and $X_1, \ldots, X_n$, it holds that:*

$$\frac{1}{n} \sum_{i=1}^n \mathbb{E}\left[ (\mu_i - \hat{\mu}_i^{EBCF})^2 \mid X_{1:n}, \mu_{1:n} \right] < \frac{1}{n} \sum_{i=1}^n \mathbb{E}\left[ (\mu_i - Z_i)^2 \mid X_{1:n}, \mu_{1:n} \right] = \sigma^2$$

**Non-Gaussian data with equal variance** Next we drop the Gaussianity assumption, and consider the model (9) in full generality. We use properties of SURE outlined above to establish the following:

**Theorem 7.** *Assume the pairs $(X_i, Z_i)_{1 \leq i \leq n}$ are independent and satisfy (9). Furthermore assume that there exist $\Gamma, M < \infty$ such that $\sup_i \mathbb{E}\left[ Z_i^4 \mid \mu_i, X_i \right] \leq \Gamma^4$ and that $\sup_i |\mu_i| \leq M$, $\sup_x |\hat{m}_{I_1}(x)| \leq M$ almost surely. Then (the analogous claim holds also with $I_1, I_2$ flipped):*

$$\sup_{A \geq 0} \left\{ \frac{1}{|I_2|} \sum_{i \in I_2} \mathbb{E}\left[ \left( \mu_i - \hat{\mu}_i^{EBCF} \right)^2 - \left( \mu_i - t^*_{\hat{m}_{I_1}, A}(X_i, Z_i) \right)^2 \mid X_{1:n}, \mu_{1:n}, Z_{I_1} \right] \right\} \leq O\left( \frac{1}{\sqrt{|I_2|}} \right)$$

**Corollary 8.** *Assume that $|I_1| = |I_2| = n/2$ and $(X_i, \mu_i, Z_i)$ are i.i.d. and satisfy the assumptions of Theorem 7. Then, the following holds, with $(X, \mu)$ a fresh draw from (9):*

$$\frac{1}{n} \sum_{i=1}^n \mathbb{E}\left[ \left( \mu_i - \hat{\mu}_i^{EBCF} \right)^2 \right] \leq \frac{\sigma^2 \mathbb{E}\left[ \left( \hat{m}_{n/2}(X) - \mu \right)^2 \right]}{\sigma^2 + \mathbb{E}\left[ \left( \hat{m}_{n/2}(X) - \mu \right)^2 \right]} + O\left( \frac{1}{\sqrt{n}} \right). \tag{12}$$

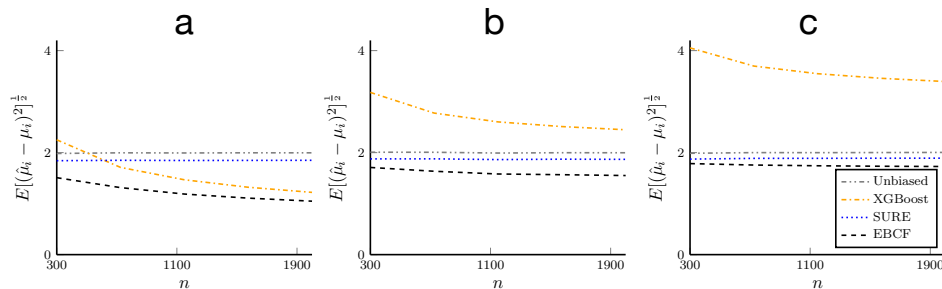

Figure 2: **Root mean squared error (RMSE) for estimating $\mu_i$ in model** (1). Results are shown as a function of $n$ for the four estimators described in the main text. **a)** Here we let $\sigma = 2$, $A = 0$ corresponding to the case of nonparametric regression. In panel **b)**, we let $\sigma = \sqrt{A} = 2.0$ corresponding to intermediate shrinkage and in panel **c)** we let $\sigma = 2$, $\sqrt{A} = 3$. The standard errors of all RMSEs are smaller or equal to 0.01.

Here $\hat{m}_{n/2}(\cdot)$ is the fitted function based on $n/2$ samples $(X_i, Z_i)$. To interpret this result, we note that when $\hat{m}(\cdot)$ can accurately capture $\mu_i$, i.e., $\hat{m}(\cdot)$ is a good estimate of $m(\cdot)$ and $\mu_i$ can be well explained using the available covariates $X_i$, the error in (12) essentially matches the error of the direct regression-based method $\hat{\mu}_i := \hat{m}_{n/2}(X_i)$. Conversely, when the error of $\hat{m}(\cdot)$ for estimating $\mu_i$ is large, we recover the error $\sigma^2$ of the simple estimator $\hat{\mu}_i := Z_i$. But in the interesting regime where the mean-squared error of $\hat{m}(\cdot)$ for $\mu_i$ is comparable to $\sigma^2$, we can do a much better job by taking a convex combination of the regression prediction $\hat{m}_{n/2}(X_i)$ and $Z_i$, and the EBCF estimator automatically and robustly navigates this trade-off.

**Non-Gaussian data with unequal variance:**  Finally, we note that we may even drop the assumption of equal variance and assume each unit has its own (conditional) variance $\sigma_i^2$ in (9) rather than the same $\sigma^2$ for everyone. We may think of the Bayes estimator (2) as also being a function of $\sigma_i$, i.e. write it as $t_{m,A}^*(x, z, \sigma)$. Then, the EBCF estimator takes the following form: For $i \in I_2$ we estimate $\mu_i$ by $t_{\hat{m}_{I_1}, \hat{A}_{I_2}}^*(X_i, Z_i, \sigma_i)$. We get $\hat{m}_{I_1}$ by regression, while for $\hat{A}_{I_2}$, we generalize (10):

$$\hat{A}_{I_2} = \operatorname*{argmin}_{A \geq 0} \left\{ \mathrm{SURE}_{I_2}(A) \right\}, \ \mathrm{SURE}_{I_2}(A) = \frac{1}{|I_2|} \sum_{i \in I_2} \left( \sigma_i^2 + \frac{\sigma_i^4}{(A + \sigma_i^2)^2}(Z_i - \hat{m}_{I_1}(X_i))^2 - 2\frac{\sigma_i^4}{A + \sigma_i^2} \right)$$

The result of Theorem 7 (see Appendix C.2) also holds in this case and we demonstrate the claims in the empirical application on the MovieLens dataset below.

## 5   Empirical results

For our empirical results we compare the following 4 estimation methods for $\mu_i$: **a)** The unbiased estimator $\hat{\mu}_i := Z_i$, **b)** the out-of-fold [4] regression prediction $\hat{\mu}_i := \hat{m}(X_i)$, where $\hat{m}$ is the fit from boosted regression trees, as implemented in XGBoost [Chen and Guestrin, 2016] with number of iterations chosen by 5-fold cross-validation and $\eta = 0.1$ (weight with which new trees are added to the ensemble), **c)** the empirical Bayes estimator (2) without covariates that shrinks $Z_i$ towards the grand average $\sum_{i=1}^n Z_i/n$, with tuning parameters selected via SURE following [Xie et al., 2012], and **d)** the proposed EBCF (empirical Bayes with cross-fitting) method, with 5 folds used for cross-fitting and XGBoost as the regression learner (with cross-validation nested within cross-fitting).

**Synthetic data**: We generate data from model (1) with $\mathbb{P}^X = U[0, 1]^{15}$ and $m(\cdot)$ is the Friedman [1991] function $m(x) = 10\sin(\pi x_1 x_2) + 20(x_3 - 1/2)^2 + 10x_4 + 5x_5$, and the last 10 coordinates are noise. Furthermore, we let $\sigma = 2.0$ and vary $A \in \{0, 4, 9\}$, mimicking the three cases in Figure 1, and we also vary $n$. Results are averaged over 100 simulations and shown in Figure 2. We make the following observation: The unbiased estimator $Z_i$ and the SURE estimator which shrinks towards the grand mean have constant mean squared error and results do not improve with increasing $n$. The XGBoost predictor improves with increasing $n$, since $m(\cdot)$ is estimated more accurately; indeed in panel a), if $\hat{m}(\cdot)$ would be exactly equal to $m(\cdot)$, then the error would be 0. However, as seen in panels

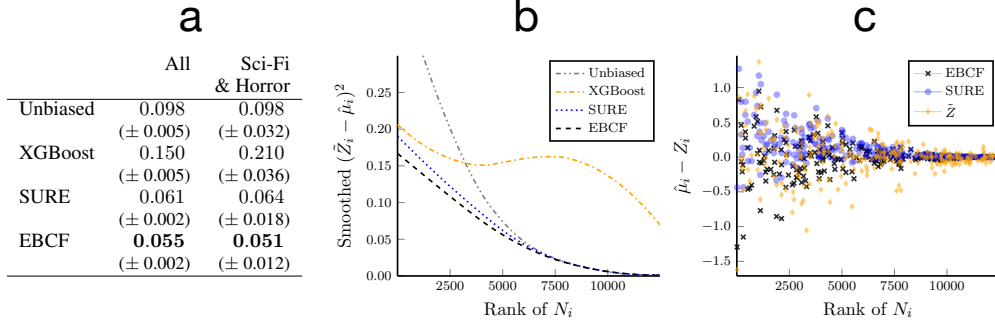

Figure 3: **EB analysis of the Movielens dataset for prediction of average movie rating. a)** Mean-squared error (MSE) $n^{-1} \sum_{i=1}^{n} (\hat{\mu}_i - \tilde{Z}_i)^2$ ($\pm$ 2 standard errors of the MSE ) of four estimators for the Movielens dataset (where $\tilde{Z}_i$ is the average rating computed from the heldout data with 90% of users) for all movies, as well as the subset of movies that are classified as both Horror and Sci-Fi. **b)** LOESS smooth of mean squared error across all movies against the rank of $N_i$, where $N_i$ is the number of users that rated movie $i$ in the training set. **c)** Deviations of EBCF (empirical Bayes with cross-fitting) and SURE (Stein's unbiased risk estimate) predictions from the unbiased estimator $Z_i$ as a function of $N_i$ for all Horror & Sci-Fi movies. We also show the "true" errors $\tilde{Z}_i - Z_i$.

$b, c$), when $A > 0$, the mean squared error of XGBoost is lower bounded by $A$, even under perfect prediction of $m(\cdot)$. In contrast, EBCF always improves with $n$ by leveraging the improved predictions of XGBoost, and outperforms all other estimators, even in the case $A = 0$ which corresponds to nonparametric regression.

**MovieLens data** [Harper and Konstan, 2016]: Here we elaborate on the example from the introduction which aims to predict the average movie rating given ratings from a finite number of users. The MovieLens dataset consists of approximately 20 million ratings in $\{0, 0.5, \ldots, 5\}$ from 138,000 users applied to 27,000 movies. To demonstrate the applicability of our approach, when model (1) does not necessarily hold, we randomly choose 10% of all users and attempt to estimate the movie ratings from them. This corresponds to having a much smaller dataset. We then summarize the $i$-th movie, by $Z_i$, the average of the $N_i$ users (in the training dataset) that rated it. We further have covariates $X_i \in \mathbb{R}^{20}$ that include $N_i$, the year the movie was released, as well as indicators of 18 genres to which the movie may belong (action, comedy, etc.). We posit that $Z_i \perp \mu_i, X_i \sim (\mu_i, \sigma^2/N_i)$ and want to estimate $\mu_i$.[5] As our pseudo ground truth for movie $i$ we use $\tilde{Z}_i$, the average movie rating among the remaining 90% of users and then report the error $\sum_{i=1}^{n} (\tilde{Z}_i - \hat{\mu}_i)^2/n$, where $n$ is the total number of movies.[6]

The average error across all movies is shown in Figure 3a; here the XGBoost predictor performs worst, followed by the unbiased estimator $Z_i$. Instead, the two EB approaches perform a lot better with EBCF scoring the lowest error. The same is true when comparing only the 253 movies with genre tags for both horror and Sci-Fi. In panel b), we show the relationship between the error $(\tilde{Z}_i - \hat{\mu}_i)^2$ and the rank of the per-movie number of reviews $N_i$ using a LOESS smoother [Cleveland and Devlin, 1988]. We observe that the 3 estimators that use $Z_i$, do a perfect job for large $N_i$ and a worse job for smaller $N_i$. In particular, the error of $Z_i$ blows up at small $N_i$, and the error gains of EBCF occur precisely at low sample sizes. On the other hand, the XGBoost prediction has an error that does not get reduced by larger $N$, but is competitive at small $N$. Panel c) shows $\hat{\mu}_i - Z_i$ for the 253 predictions of EBCF and SURE for horror/Sci-Fi movies as a function of the rank of $N_i$. For large $N_i$, again both EB estimators agree with the unbiased estimator. However, for small $N_i$, it appears that most Sci-Fi/Horror movies are worse than the average movie, and EB without covariates tries to correct for this by assigning them a higher rating. Conversely, EBCF automatically realizes that these movies tend to get low ratings, and pulls the unbiased estimator $Z_i$ further down.

**Communities and Crimes data** from the UCI repository [Dua and Graff, 2017, Redmond and Baveja, 2002]: The dataset provides information about the number of crimes in multiple US communities

| | $B = 200$ MSE $(\times 10^6)$ | $B = 500$ MSE $(\times 10^6)$ |
|---|---|---|
| Unbiased | $223.9\ (\pm 16.8)$ | $92.2\ (\pm 7.1)$ |
| XGBoost | $398.0\ (\pm 81.8)$ | $370.2\ (\pm 78.6)$ |
| SURE | $184.2\ (\pm 18.9)$ | $85.6\ (\pm 7.2)$ |
| EBCF | $\mathbf{152.0}\ (\pm 22.2)$ | $\mathbf{78.5}\ (\pm 10.3)$ |

Table 1: **EB analysis of the Communities and Crimes dataset:** The table reports the mean-squared error ($\pm$ 2 standard errors) of four different estimators for the non-violent crime rate. The columns correspond to down-sampling the dataset to a population of $B = 200$ or $B = 500$ for each community.

as compiled by the FBI Uniform Crime Reporting program in 1995. Our task is to predict the non-violent crime rate $p_i$ of community $i$, defined as $p_i :=$ Crimes in community $i$/Population $i$, for each of $n = 2118$ communities[7]. We observe a dataset in which the population of each community is down-sampled to $B = 200$ as

$$C_i \sim \text{Hypergeometric}(B,\ \text{Crimes in community } i,\ \text{Population } i)$$

We seek to predict $p_i$ based on $C_i$ and covariates $X_i \in \mathbb{R}^{74}$ which include all unnormalized, numeric predictive covariates in the UCI data set description (after removing covariates with missing entries) and comprise features derived from Census and law enforcement data, such as percentage of people that are employed and percentage of police officers assigned to drug units. We note that the hypergeometric subsampling makes the estimation task harder and also provides us with pseudo ground truth $p_i$; cf. Wager [2015] for further motivation of such down-sampling.

The problem may be cast into the setting of this paper by defining $Z_i := \sqrt{C_i/B}$. Then, by a variance stabilizing argument, it follows that $Z_i \overset{.}{\sim} \left(\sqrt{p_i}, 1/(4 \cdot B)\right)$ and we may apply the same methods as in the preceding examples to estimate $\mu_i := \sqrt{p_i}$ by $\hat{\mu}_i$. After transforming the estimates back to the original scale through $\hat{p}_i = \hat{\mu}_i^2$, we report the error $\sum_{i=1}^{n}(p_i - \hat{p}_i)^2/n$, where $n$ is the number of communities analyzed. Table 1 shows the results of this analysis, as well as the same analysis repeated for $B = 500$. EBCF shows promising performance compared to the other baselines for both $B$. As we decrease the amount of downsampling from $B = 200$ to $B = 500$, we see that methods that depend on $Z_i$ (unbiased, SURE and EBCF) improve a lot, while XGBoost does not.

## 6 Discussion

Empirical Bayes is a powerful framework for pooling information across many experiments, and improve the precision of our inference about each experiment on its own [Efron, 2010, Robbins, 1964]. Existing empirical Bayes methods, however, do not allow the analyst to leverage covariate information unless they assume a rigid parametric model as in Fay and Herriot [1979], or are willing to commit to a specific end-to-end estimation strategy as in, e.g., Opsomer et al. [2008]. In contrast, the approach proposed here allows the analyst to perform covariate-powered empirical Bayes estimation on the basis of any black-box predictive model, and has strong formal properties whether or not the model (1) used to motivate our procedure is well specified. Our approach may be extended in future work by considering generalizations of (1), such as covariate-based modulation of the prior variance, i.e., $\mu_i \mid X_i \sim \mathcal{N}(m(X_i), A(X_i))$. The working assumption of a normal prior could also be replaced by heavy-tailed priors [Zhu, Ibrahim, and Love, 2018] or priors with a point mass at zero.

The prevalence of settings where we need to analyze results from many loosely related experiments seems only destined to grow, and we believe that empirical Bayes methods that allow for various forms of structured side information hold promise for fruitful application across several different areas.

**Code availability and reproducibility**
The proposed EBCF (empirical Bayes with cross-fitting) method has been implemented in EBayes.jl (https://github.com/nignatiadis/EBayes.jl), a package written in the Julia language [Bezanson et al., 2017]. Dependencies of EBayes.jl include MLJ.jl [Blaom et al., 2019], Optim.jl [Mogensen and Riseth, 2018] and Distributions.jl [Besançon et al., 2019]. We also provide a Github repository (https://github.com/nignatiadis/EBCrossFitPaper) with code to reproduce all empirical results in this paper, including a specification for downloading the dependencies and datasets.

**Acknowledgments**

The authors are grateful for enlightening conversations with Brad Efron, Guido Imbens, Panagiotis Lolas and Paris Syminelakis. This research was funded by a gift from Google.

## Footnotes

[1]We will propose procedures adaptive to unknown $A$ in Section 3.

[2]Throughout, we use the following notation for the asymptotic rates: For two sequences $a_n, b_n > 0$, we say $a_n \lesssim b_n$ if $\limsup_{n \to \infty} a_n/b_n \leq c$ for a constant $c$ that *does not* depend on $A, \sigma, n$. Similarly, we say $a_n \gtrsim b_n$ if $b_n \lesssim a_n$ and finally $a_n \asymp b_n$ if both $a_n \gtrsim b_n$ and $a_n \lesssim b_n$.

[3]Li and Hwang [1984] provide a similar result when $\hat{m}(\cdot)$ is a linear smoother.

[4]By out-of-fold we mean that the regression prediction is the one used by 5-fold EBCF described below.

[5]We replace $\sigma^2$ by $\hat{\sigma}^2 \doteq 0.94$, the average of the sample standard deviations across all movies.

[6]We filter movies and keep only movies with at least 3 ratings in the training set and 11 in the validation set.

[7]We filter out communities with a missing number of non-violent crimes.

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
