[Supplementary Material]

# A   Proofs for Section 2

## A.1   Proof of Theorem 2

*Proof.* We will first show, that under model (1), the plug-in estimator (7) satisfies:

$$\mathbb{E}_{m,A}\left[L(\hat{t}_n; m, A)\right] = \frac{\sigma^4}{\left(\sigma^2 + A\right)^2}\mathbb{E}_{m,A}\left[(\hat{m}_n(X_{n+1}) - m(X_{n+1}))^2\right] \tag{13}$$

This also establishes the upper bound on the minimax excess risk if $\hat{m}_n$ is chosen in a minimax rate-optimal way for the regression problem.

To prove (13), we study the excess risk of this estimator conditionally on the covariate $X_{n+1}$ of the $n+1$-th observation:

$$\mathbb{E}_{m,A}\left[\left(\hat{t}(X_{n+1}, Z_{n+1}) - \mu_{n+1}\right)^2 \mid X_{n+1} = x\right]$$

$$= \mathbb{E}_{m,A}\left[\left(\frac{A}{\sigma^2 + A}Z_{n+1} + \frac{\sigma^2}{\sigma^2 + A}\hat{m}(X_{n+1}) - \mu_{n+1}\right)^2 \mid X_{n+1} = x\right]$$

$$= \mathbb{E}_{m,A}\left[\left(\frac{A}{\sigma^2 + A}Z_{n+1} + \frac{\sigma^2}{\sigma^2 + A}m(X_{n+1}) - \mu_{n+1} + \frac{\sigma^2}{\sigma^2 + A}\left(\hat{m}(X_{n+1}) - m(X_{n+1})\right)\right)^2 \mid X_{n+1} = x\right]$$

$$= \mathbb{E}_{m,A}\left[\left(t_{m,A}^*(X_{n+1}, Z_{n+1}) - \mu_{n+1}\right)^2 \mid X_{n+1} = x\right] + \frac{\sigma^4}{\left(\sigma^2 + A\right)^2}\mathbb{E}_{m,A}\left[(\hat{m}(X_{n+1}) - m(X_{n+1}))^2 \mid X_{n+1} = x\right]$$

The result follows by integrating over $X_{n+1}$ and rearranging. $\qquad\square$

## A.2   Proof of Lemma 1

The idea of the proof follows the general paradigm in derivation of minimax optimal rates [Tsybakov, 2008, Duchi, 2019] in which we reduce the original problem to a multiple hypothesis testing problem. More concretely, let us fix two functions $m_1, m_2 \in \mathcal{C}$ and call the induced distributions $P_1, P_2$. Say we have a denoiser $t(x, z)$ that performs extremely well under $m_1$ with respect to the loss (3). Then we will argue that it cannot do too well under $m_2$. But then, given data $(X_1, Z_1), \ldots, (X_n, Z_n)$ we may use the data-driven $\hat{t}(x, z)$ as a proxy for a hypothesis test: If its risk is small under $m_1$, but large under $m_2$, we would guess that $m_1$ is true and vice versa. Thus our task reduces to lower bounding the performance of a hypothesis test. These ideas will be made concrete in the arguments that follow.

Our proof strategy begins by studying the pointwise excess risk:

$$L(t; m, A \mid x) := \mathbb{E}_{m,A}\left[(t(x, Z_{n+1}) - \mu_{n+1})^2 - \left(t_{m,A}^*(x, Z_{n+1}) - \mu_{n+1}\right)^2 \mid X_{n+1} = x\right] \tag{14}$$

**Lemma 9.** *There exist universal constants $c > 0, \Delta > 0$ such that when $|m_1(x) - m_2(x)|/\sqrt{\sigma^2 + A} \le \Delta$ (where $x$ is fixed, yet arbitrary) it holds for all $t$ that:*

$$\frac{1}{2}\left[L(t; m_1, A \mid x) + L(t; m_2, A \mid x)\right] \ge c\frac{\sigma^4}{\left(\sigma^2 + A\right)^2}\left(m_1(x) - m_2(x)\right)^2$$

*Proof.* As a thought experiment, we consider the following generative model:

$$\mu_{n+1} \sim G_x = \frac{1}{2}\left[\mathcal{N}\left(m_1(x), A\right) + \mathcal{N}\left(m_2(x), A\right)\right]$$

$$Z_{n+1} \mid \mu_{n+1} \sim \mathcal{N}\left(\mu_{n+1}, \sigma^2\right)$$

Next consider the Bayes estimator for $\mu_{n+1}$ under this prior, namely:

$$t_{G_x}^*(z) := \mathbb{E}_{G_x}\left[\mu_{n+1} \mid Z_{n+1} = z\right] \tag{15}$$

Then, by definition of the Bayes estimator, it must hold that for any $t : \mathcal{X} \times \mathbb{R} \to \mathbb{R}$:

$$\mathbb{E}_{G_x}\left[\left(t(x, Z_{n+1}) - \mu_{n+1}\right)^2\right] \geq \mathbb{E}_{G_x}\left[\left(t^*_{G_x}(Z_{n+1}) - \mu_{n+1}\right)^2\right]$$

In the preceding result we are really thinking of $t$ as the curried function $t(x, \cdot)$. Next, by definition of $G_x$, the LHS of the above expression is the same as:

$$\frac{1}{2}\left\{\mathbb{E}_{m_1, A}\left[\left(t(x, Z_{n+1}) - \mu_{n+1}\right)^2 \mid X_{n+1} = x\right] + \mathbb{E}_{m_2, A}\left[\left(t(x, Z_{n+1}) - \mu_{n+1}\right)^2 \mid X_{n+1} = x\right]\right\}$$

Also observe that $\inf_t \left\{\mathbb{E}_{m_1, A}\left[\left(t(x, Z_{n+1}) - \mu_{n+1}\right)^2 \mid X_{n+1} = x\right]\right\} = A\sigma^2/(A + \sigma^2)$ and similarly for $m_2$, hence upon subtracting $A\sigma^2/(A + \sigma^2)$ from the above expression and its preceding inequality, we get:

$$\frac{1}{2}\left\{L(t; m_1, A \mid x) + L(t; m_2, A \mid x)\right\} \geq \mathbb{E}_{G_x}\left[\left(t^*_{G_x}(Z_{n+1}) - \mu_{n+1}\right)^2\right] - \frac{A\sigma^2}{A + \sigma^2}$$

Hence to conclude we will need to show that there exist universal constants $c, \Delta > 0$ so that if $|m_1(x) - m_2(x)|/\sqrt{\sigma^2 + A} \leq \Delta$:

$$\mathbb{E}_{G_x}\left[\left(t^*_{G_x}(Z_{n+1}) - \mu_{n+1}\right)^2\right] - \frac{A\sigma^2}{A + \sigma^2} \geq c\frac{\sigma^4}{(\sigma^2 + A)^2}\left(m_1(x) - m_2(x)\right)^2 \qquad (16)$$

Note that the LHS depends on $m_1(x), m_2(x)$ through the definition of $G_x$. We provide the calculations and complete the proof in Appendix A.3. $\qquad\square$

**Lemma 10.** *Let $c > 0$, $\Delta > 0$ the constants from Lemma 9. Then, for all $m_1, m_2 : \mathcal{X} \to \mathbb{R}$, the following implication holds for any $t : \mathcal{X} \times \mathbb{R} \to \mathbb{R}$*

$$L(t; m_1, A) < c\frac{\sigma^4}{(\sigma^2 + A)^2}\int (m_1(x) - m_2(x))^2 \, \mathbf{1}\left\{\frac{(m_1(x) - m_2(x))^2}{\sigma^2 + A} \leq \Delta^2\right\} d\mathbb{P}^X(x)$$

$$\implies L(t; m_2, A) \geq c\frac{\sigma^4}{(\sigma^2 + A)^2}\int (m_1(x) - m_2(x))^2 \, \mathbf{1}\left\{\frac{(m_1(x) - m_2(x))^2}{\sigma^2 + A} \leq \Delta^2\right\} d\mathbb{P}^X(x)$$

$$(17)$$

*Proof.* We use the result from Lemma (9), noting that $L(t; m, A) = \int L(t; m, A \mid x) d\mathbb{P}^X(x)$.

$$\frac{1}{2}\left[L(t; m_1, A) + L(t; m_2, A)\right] = \int \frac{1}{2}\left[L(t; m_1, A \mid x) + L(t; m_2, A \mid x)\right] d\mathbb{P}^X(x)$$

$$\geq \int \frac{1}{2}\left[L(t; m_1, A \mid x) + L(t; m_2, A \mid x)\right] \mathbf{1}\left\{\frac{(m_1(x) - m_2(x))^2}{\sigma^2 + A} \leq \Delta^2\right\} d\mathbb{P}^X(x)$$

$$\geq c\frac{\sigma^4}{(\sigma^2 + A)^2}\int (m_1(x) - m_2(x))^2 \, \mathbf{1}\left\{\frac{(m_1(x) - m_2(x))^2}{\sigma^2 + A} \leq \Delta^2\right\} d\mathbb{P}^X(x)$$

Thus not both $L(t; m_1, A), L(t; m_2, A)$ may be $<$ than the RHS at the same time.

$\qquad\square$

The above lemma allows us to prove lower bounds by reduction to hypothesis testing. In particular, let us recall the statement from Lemma 1, now stated in slightly more generality and dropping explicit notation for $n$ in the constructed collection of functions $\{m_v \mid v \in \mathcal{V}_n\}$:

**Lemma 1 (More general version).** *For each $n$, let $\mathcal{V}_n$ be a finite set and $\{m_v \mid v \in \mathcal{V}_n\} \subset \mathcal{C}$ be a collection of functions indexed by $\mathcal{V}_n$ such that for a sequence $\delta_n > 0$:*

$$\delta_n^2 \leq \int \left(m_v(x) - m_{v'}(x)\right)^2 \mathbf{1} \left\{ \frac{\left(m_v(x) - m_{v'}(x)\right)^2}{\sigma^2 + A} \leq \Delta^2 \right\} d\mathbb{P}^X(x) \ \textit{for all } v \neq v' \in \mathcal{V}_n, \forall n$$

*Then:*

$$\mathfrak{M}_n^{EB}\left(\mathcal{C}; A, \sigma^2\right) \gtrsim \frac{\sigma^4}{\left(\sigma^2 + A\right)^2} \cdot \delta_n^2 \cdot \inf_{\hat{V}_n} \mathbb{P}\left[\hat{V}_n \neq V_n\right]$$

*Here,* $\inf_{\hat{V}_n} \mathbb{P}[\hat{V}_n \neq V_n]$ *is to be interpreted as follows:* $V_n$ *is drawn uniformly from* $\mathcal{V}_n$ *and condition-ally on* $V_n = v$, *we draw the pairs* $(X_i, Z_i)_{1 \leq i \leq n}$ *from model* (1) *with regression function* $m_{n,v}(\cdot)$. *The infimum is taken over all estimators* $\hat{V}_n$ *that are measurable with respect to* $(X_i, Z_i)_{1 \leq i \leq n}$.

Note that the original statement of Lemma 1 is subsumed by the above statement. We are ready to prove Lemma 1.

*Proof.* Our construction closely follows Duchi [2019] and recent advances in proving minimax results for general losses; see for example [Agarwal et al., 2009]. To start, we fix an estimated denoiser $\hat{t}_n(x, z) = \hat{t}_n\left(x, z; (X_i, Z_i)_{1 \leq i \leq n}\right)$ and define $\delta_{n,A,\sigma} = c^{1/2} \frac{\sigma^2}{\sigma^2 + A} \delta_n$, where $c$ is defined in Lemma 9. Next, focusing on one $v \in \mathcal{V}$, we get by Markov's inequality:

$$\mathbb{E}_{m_v,A}\left[L(\hat{t}_n; m_v, A)\right] \geq \delta_{n,A,\sigma}^2 \mathbb{P}_{m_v,A}\left[L(\hat{t}_n; m_v, A) \geq \delta_{n,A,\sigma}^2\right]$$

We next construct an estimator $\tilde{V}_n$ of $V_n$, namely we let:

$$\tilde{V}_n = \text{argmin}_{v \in \mathcal{V}} L(\hat{t}_n; m_v, A)$$

Notice that by Lemma 10 and the assumption of the current Lemma, if the truth is $m_v$ and $L(\hat{t}_n; m_v, A) < \delta_{n,A,\sigma}^2$, then we definitely guessed correctly, in other words:

$$L(\hat{t}_n; m_v, A) < \delta_{n,A,\sigma}^2 \implies \tilde{V}_n = v$$

But taking the complements:

$$\tilde{V}_n \neq v \implies L(\hat{t}_n; m_v, A) \geq \delta_{n,A,\sigma}^2$$

In terms of probabilities this implies that

$$\mathbb{P}_{m_v,A}\left[L(\hat{t}_n; m_v, A) \geq \delta_{n,A,\sigma}^2\right] \geq \mathbb{P}_{m_v,A}\left[\tilde{V}_n \neq v\right]$$

Combining with our original result, and averaging over all $v$, we see that:

$$\begin{aligned}
\sup_{m \in \mathcal{C}} \left\{\mathbb{E}_m\left[L(\hat{t}_n; m, A)\right]\right\} &\geq \sup_{v \in \mathcal{V}_n} \left\{\mathbb{E}_{m_v,A}\left[L(\hat{t}_n; m_v, A)\right]\right\} \\
&\geq \frac{1}{|\mathcal{V}_n|} \sum_{v \in \mathcal{V}_n} \mathbb{E}_{m_v,A}\left[L(\hat{t}_n; m_v, A)\right] \\
&\geq \delta_{n,A,\sigma}^2 \left\{\frac{1}{|\mathcal{V}_n|} \sum_{v \in \mathcal{V}_n} P_{m_v,A}\left[\tilde{V}_n \neq v\right]\right\} \\
&= \delta_{n,A,\sigma}^2 \mathbb{P}\left[\tilde{V}_n \neq V_n\right] \\
&\geq \delta_{n,A,\sigma}^2 \inf_{\hat{V}_n} \mathbb{P}\left[\hat{V}_n \neq V_n\right]
\end{aligned}$$

Recall the definition of $\delta_{n,A,\sigma}^2$ and that $\hat{t}_n$ was arbitrary to conclude. $\square$

### A.3 Proof of Lemma 9

*Proof.* It only remains to prove (16). To this end, let us note that the result is essentially univariate; i.e. we may consider the following model:

$$
\begin{aligned}
\mu &\sim\ G = \frac{1}{2}\left[\mathcal{N}\left(\eta_1, A\right) + \mathcal{N}\left(\eta_2, A\right)\right] \\
Z \mid \mu &\sim \mathcal{N}\left(\mu, \sigma^2\right)
\end{aligned}
\tag{18}
$$

In this model, we want to prove that the Bayes risk of the Bayes estimator $t_G^*(Z) = \mathbb{E}_G\left[\mu \mid Z\right]$ satisfies the following inequality ($c > 0, \Delta > 0$): When $|\eta_1 - \eta_2| \le \Delta\sqrt{\sigma^2 + A}$ it holds that

$$
\mathbb{E}_G\left[(t_G^*(Z) - \mu)^2\right] - \frac{A\sigma^2}{A + \sigma^2} \ge c\frac{\sigma^4}{(\sigma^2 + A)^2}(\eta_1 - \eta_2)^2
\tag{19}
$$

The calculation is facilitated by Lemma 11, which states that $\mathbb{E}_G\left[(t_G^*(Z) - \mu)^2\right] = \sigma^2\left[1 - \sigma^2 I(f_g)\right]$, where $f_g$ is the marginal density of $Z$ in (18) and $I(f_g)$ is the Fisher information $\int \frac{f_g'(x)^2}{f_g(x)}dx$.

For the problem at hand, without loss of generality, we may take $\eta_1 = 0, \eta_2 = \eta > 0$. Then the marginal distribution induced by $g$ is the mixture $\frac{1}{2}\left[\mathcal{N}\left(0, \sigma^2 + A\right) + \mathcal{N}\left(\eta, \sigma^2 + A\right)\right]$, i.e. the pdf $f_g(\cdot)$ is:

$$
f_g(x) = \frac{1}{2\sqrt{2\pi\left(\sigma^2 + A\right)}}\left[\exp\left(-\frac{x^2}{2(\sigma^2 + A)}\right) + \exp\left(-\frac{(x - \eta)^2}{2(\sigma^2 + A)}\right)\right]
$$

$$
f_g'(x) = \frac{1}{2\sqrt{2\pi\left(\sigma^2 + A\right)}}\frac{1}{\sigma^2 + A}\left[-x\exp\left(-\frac{x^2}{2(\sigma^2 + A)}\right) - (x - \eta)\exp\left(-\frac{(x - \eta)^2}{2(\sigma^2 + A)}\right)\right]
$$

Therefore, letting $\ell(u) = \exp(u)/(1 + \exp(u))$ the logistic function, we see that:,

$$
\frac{f_g'(x)}{f_g(x)} = \frac{1}{\sigma^2 + A}\frac{-x - (x - \eta)\exp\left(\frac{-\eta^2 + 2\eta x}{2(\sigma^2 + A)}\right)}{1 + \exp\left(\frac{-\eta^2 + 2\eta x}{2(\sigma^2 + A)}\right)} = \frac{1}{\sigma^2 + A}\left[-x + \eta\cdot\ell\left(\frac{-\eta^2 + 2\eta x}{2(\sigma^2 + A)}\right)\right]
$$

Thus:

$$
\frac{f_g'(x)^2}{f_g(x)^2} = \frac{1}{(\sigma^2 + A)^2}\left[x^2 + \eta^2\cdot\ell^2\left(\frac{-\eta^2 + 2\eta x}{2(\sigma^2 + A)}\right) - 2x\eta\cdot\ell\left(\frac{-\eta^2 + 2\eta x}{2(\sigma^2 + A)}\right)\right]
$$

Then, letting $\tilde{x} = x/\sqrt{A + \sigma^2}, \tilde{\eta} = \eta/\sqrt{A + \sigma^2}$:

$$
\begin{aligned}
I(f_g) &= \frac{1}{(\sigma^2 + A)^2}\frac{1}{2\sqrt{2\pi\left(\sigma^2 + A\right)}}\int\left\{x^2 + \eta^2\cdot\ell^2\left(\frac{-\eta^2 + 2\eta x}{2(\sigma^2 + A)}\right) - 2x\eta\cdot\ell\left(\frac{-\eta^2 + 2\eta x}{2(\sigma^2 + A)}\right)\right\} \\
&\qquad\qquad\qquad\cdot\left(\exp\left(-\frac{x^2}{2(\sigma^2 + A)}\right) + \exp\left(-\frac{(x - \eta)^2}{2(\sigma^2 + A)}\right)\right)dx \\
&= \frac{1}{2\sqrt{2\pi}\left(\sigma^2 + A\right)}\int\left\{\tilde{x}^2 + \tilde{\eta}^2\cdot\ell^2\left(\frac{-\tilde{\eta}^2 + 2\tilde{\eta}\tilde{x}}{2}\right) - 2\tilde{x}\tilde{\eta}\cdot\ell\left(\frac{-\tilde{\eta}^2 + 2\tilde{\eta}\tilde{x}}{2}\right)\right\} \\
&\qquad\qquad\qquad\cdot\left(\exp\left(-\frac{\tilde{x}^2}{2}\right) + \exp\left(-\frac{(\tilde{x} - \tilde{\eta})^2}{2}\right)\right)d\tilde{x}
\end{aligned}
$$

Thus we may write $I(f_g) = \frac{1}{\sigma^2 + A}C(\tilde{\eta})$, for some $C(\tilde{\eta})$, which we now turn to study. Our first observation is that $C(0) = \mathbb{E}\left[\widetilde{X}^2\right] = 1$ where $\widetilde{X} \sim \mathcal{N}(0, 1)$. We claim that:

$$C(\tilde{\eta}) = 1 - \frac{\tilde{\eta}^2}{4} + o(\tilde{\eta}^2)$$

To this end, we break up $C(\eta)$ into 6 components upon distributing terms, calling them $\mathrm{I}_0, \mathrm{II}_0, \mathrm{III}_0, \mathrm{I}_{\tilde{\eta}}, \mathrm{II}_{\tilde{\eta}}, \mathrm{III}_{\tilde{\eta}}$, where the subscript corresponds to integrating over $\widetilde{X} \sim \mathcal{N}(0,1)$ or $\widetilde{X} \sim \mathcal{N}(\tilde{\eta}, 1)$.

$$\mathrm{I}_0 := \mathbb{E}_0\left[\widetilde{X}^2\right] = 1, \quad \mathrm{I}_{\tilde{\eta}} := \mathbb{E}_{\tilde{\eta}}\left[\widetilde{X}^2\right] = 1 + \tilde{\eta}^2$$

$$\mathrm{II}_0 := \mathbb{E}_0\left[\tilde{\eta}^2 \cdot \ell^2\left(\frac{-\tilde{\eta}^2 + 2\tilde{\eta}\widetilde{X}}{2}\right)\right] = \frac{\tilde{\eta}^2}{4} + o(\tilde{\eta}^2) \qquad \text{(dominated convergence theorem)}$$

$$\mathrm{II}_{\tilde{\eta}} := \mathbb{E}_{\tilde{\eta}}\left[\tilde{\eta}^2 \cdot \ell^2\left(\frac{-\tilde{\eta}^2 + 2\tilde{\eta}\widetilde{X}}{2}\right)\right] = \frac{\tilde{\eta}^2}{4} + o(\tilde{\eta}^2)$$

We may see the last result for example as follows, again using dominated convergence ($\tilde{\eta} \to 0$):

$$\frac{\mathrm{II}_{\tilde{\eta}}}{\tilde{\eta}^2} = \mathbb{E}_{\tilde{\eta}}\left[\ell^2\left(\frac{-\tilde{\eta}^2 + 2\tilde{\eta}\widetilde{X}}{2}\right)\right] = \mathbb{E}_0\left[\ell^2\left(\frac{-\tilde{\eta}^2 + 2\tilde{\eta}(\widetilde{X} + \tilde{\eta})}{2}\right)\right] = \frac{1}{4} + o(1)$$

To bound III, it will be convenient to note that by Taylor's theorem it holds that $\ell(u) = \frac{1}{2} + \frac{u}{4} + O(u^3)$; in fact $\left|\ell(u) - \frac{1}{2} - \frac{u}{4}\right| \leq |u|^3$. Thus:

$$\ell\left(\frac{-\tilde{\eta}^2 + 2\tilde{\eta}\widetilde{X}}{2}\right) = \frac{1}{2} - \frac{\tilde{\eta}^2}{8} + \frac{\tilde{\eta}\widetilde{X}}{4} + O\left((-\tilde{\eta}^2 + 2\tilde{\eta}\widetilde{X})^3\right)$$

, and so (one may check that again dominated convergence applies):

$$\mathrm{III}_0 := -2\tilde{\eta}\mathbb{E}_0\left[\widetilde{X}\ell\left(\frac{-\tilde{\eta}^2 + 2\tilde{\eta}\widetilde{X}}{2}\right)\right] = -2\tilde{\eta}\mathbb{E}_0\left[\widetilde{X}\left(\frac{1}{2} - \frac{\tilde{\eta}^2}{8} + \frac{\tilde{\eta}\widetilde{X}}{4}\right) + O\left(\widetilde{X}(-\tilde{\eta}^2 + 2\tilde{\eta}\widetilde{X})^3\right)\right]$$

$$= -2\tilde{\eta}\mathbb{E}_0\left[\widetilde{X}\left(\frac{1}{2} - \frac{\tilde{\eta}^2}{8} + \frac{\tilde{\eta}\widetilde{X}}{4}\right)\right] + o(\tilde{\eta}^2) = -2\tilde{\eta}\left(0 + \frac{\tilde{\eta}}{4}\right) + o(\tilde{\eta}^2) = -\frac{\tilde{\eta}^2}{2} + o(\tilde{\eta}^2)$$

$$\mathrm{III}_{\tilde{\eta}} := -2\tilde{\eta}\mathbb{E}_{\tilde{\eta}}\left[\widetilde{X}\ell\left(\frac{-\tilde{\eta}^2 + 2\tilde{\eta}\widetilde{X}}{2}\right)\right] = -2\tilde{\eta}\mathbb{E}_{\tilde{\eta}}\left[\widetilde{X}\left(\frac{1}{2} - \frac{\tilde{\eta}^2}{8} + \frac{\tilde{\eta}\widetilde{X}}{4}\right) + O\left(\widetilde{X}(-\tilde{\eta}^2 + 2\tilde{\eta}\widetilde{X})^3\right)\right]$$

$$= -2\tilde{\eta}\mathbb{E}_0\left[(\widetilde{X} + \tilde{\eta})\left(\frac{1}{2} - \frac{\tilde{\eta}^2}{8} + \frac{\tilde{\eta}(\widetilde{X} + \tilde{\eta})}{4}\right) + O\left((\widetilde{X} + \tilde{\eta})(-\tilde{\eta}^2 + 2\tilde{\eta}(\widetilde{X} + \tilde{\eta}))^3\right)\right]$$

$$= -2\tilde{\eta}\mathbb{E}_0\left[(\widetilde{X} + \tilde{\eta})\left(\frac{1}{2} - \frac{\tilde{\eta}^2}{8} + \frac{\tilde{\eta}(\widetilde{X} + \tilde{\eta})}{4}\right)\right] + o(\tilde{\eta}^2)$$

$$= -2\tilde{\eta}\left(\frac{\tilde{\eta}}{2} - \frac{\tilde{\eta}^3}{8} + \frac{\tilde{\eta}(\tilde{\eta}^2 + 1)}{4}\right) + o(\tilde{\eta}^2)$$

$$= -\frac{3\tilde{\eta}^2}{2} + o(\tilde{\eta}^2)$$

Add up to get :

$$C(\tilde{\eta}) = \frac{1}{2}\left[\mathrm{I}_0 + \mathrm{II}_0 + \mathrm{III}_0 + \mathrm{I}_{\tilde{\eta}} + \mathrm{II}_{\tilde{\eta}} + \mathrm{III}_{\tilde{\eta}}\right] = 1 - \frac{\tilde{\eta}^2}{4} + o(\tilde{\eta}^2)$$

Then the regret is:

$$\sigma^2 \left[ 1 - \sigma^2 I(f_g) \right] - \frac{A\sigma^2}{\sigma^2 + A} = \sigma^2 \left( 1 - \frac{\sigma^2}{\sigma^2 + A} C(\tilde{\eta}) \right) - \frac{A\sigma^2}{\sigma^2 + A}$$

$$= \sigma^2 \left[ 1 - \frac{\sigma^2}{\sigma^2 + A} \left( 1 - \frac{\tilde{\eta}^2}{4} + o(\tilde{\eta}^2) \right) \right] - \frac{A\sigma^2}{\sigma^2 + A}$$

$$= \frac{\sigma^4}{\sigma^2 + A} \frac{1}{4} \tilde{\eta}^2 + \frac{\sigma^4}{\sigma^2 + A} o(\tilde{\eta}^2)$$

$$= \frac{1}{4} \frac{\sigma^4}{(\sigma^2 + A)^2} \eta^2 + o(\tilde{\eta}^2)$$

In particular, there exist $c > 0, \Delta > 0$ such that if $\tilde{\eta} \leq \Delta$:

$$\sigma^2 \left[ 1 - \sigma^2 I(f_g) \right] - \frac{A\sigma^2}{\sigma^2 + A} \geq c \frac{\sigma^4}{(\sigma^2 + A)^2} \eta^2$$

Recalling that $\tilde{\eta} = \eta / \sqrt{A + \sigma^2}$, we conclude. We also note that we may let $c$ be arbitrarily close to $1/4$.

$\square$

## A.4   Proof and statement of Lemma 11

**Lemma 11.** *Assume $\mu \sim g$ and $Z \mid \mu \sim \mathcal{N}\left(\mu, \sigma^2\right)$. Also call $f_g$ the marginal density of $Z$ and define the Fisher information:*

$$I(f_g) := \int \frac{f_g'(x)^2}{f_g(x)} dx = \mathbb{E}_{f_g} \left[ \frac{f_g'(Z)^2}{f_g(Z)^2} \right]$$

*Then it holds that:*

$$\inf_{\hat{\mu}} \left\{ \mathbb{E}_g \left[ (\hat{\mu} - \mu)^2 \right] \right\} = \sigma^2 \left[ 1 - \sigma^2 I(f_g) \right]$$

*Remark* 12.  This formula is quite well know, see for example [Cohen, Greenshtein, and Ritov, 2013]. Mukhopadhyay and Vidakovic [1995] call it Brown's formula in light of [Brown, 1971]. We give a proof for completeness; in which we do not justify switching integration and differentiation. For our purposes we only need the result for $g$ a mixture of two normals, in which case this is valid.

*Remark* 13.  As a simple application, consider $g = \mathcal{N}(0, A)$, then $f_g = \mathcal{N}\left(0, A + \sigma^2\right)$, so that $f_g(x) = \frac{1}{\sqrt{2\pi(\sigma^2 + A)}} \exp\left( -\frac{x^2}{2(\sigma^2 + A)} \right)$ and $f_g'(x) = -\frac{1}{\sqrt{2\pi(\sigma^2 + A)}} \frac{x}{\sigma^2 + A} \exp\left( -\frac{x^2}{2(\sigma^2 + A)} \right)$. Thus $f_g'(x)^2 / f_g(x)^2 = \frac{x^2}{(\sigma^2 + A)^2}$ and $I(f_g) = \frac{1}{\sigma^2 + A}$. The above result then states:

$$\inf_{\hat{\mu}} \left\{ \mathbb{E}_g \left[ (\hat{\mu} - \mu)^2 \right] \right\} = \sigma^2 \left[ 1 - \frac{\sigma^2}{\sigma^2 + A} \right] = \frac{\sigma^2 A}{\sigma^2 + A}$$

*Proof.*  We start with noting that the Bayes estimator is given by Tweedie's [Efron, 2011] celebrated formula:

$$\mathbb{E}_g \left[ \mu \mid Z = z \right] = z + \sigma^2 \frac{f_g'(z)}{f_g(z)}$$

Then, the Bayes risk is given by (letting $\varepsilon := Z - \mu \sim \mathcal{N}\left(0, \sigma^2\right)$):

$$\inf_{\hat{\mu}} \left\{ \mathbb{E}_g \left[ (\hat{\mu} - \mu)^2 \right] \right\} = \mathbb{E}_g \left[ \left( \mu - Z - \sigma^2 \frac{f_g'(Z)}{f_g(Z)} \right)^2 \right]$$

$$= \mathbb{E}_g \left[ \left( -\varepsilon - \sigma^2 \frac{f_g'(Z)}{f_g(Z)} \right)^2 \right]$$

$$= \sigma^2 + \sigma^4 I(f_g) + 2\sigma^2 \mathbb{E}_g \left[ \varepsilon \frac{f_g'(Z)}{f_g(Z)} \right]$$

$$= \sigma^2 - \sigma^4 I(f_g)$$

It remains to justify that: $\mathbb{E}_g\left[\varepsilon \frac{f_g'(Z)}{f_g(Z)}\right] = -\sigma^2 I(f_g)$. To this end, first note that upon conditioning on $\mu$, we may use Stein's lemma, as follows:

$$
\begin{aligned}
\mathbb{E}_g\left[\varepsilon \frac{f_g'(Z)}{f_g(Z)}\right] &= \mathbb{E}_g\left[\mathbb{E}\left[\varepsilon \frac{f_g'(\varepsilon+\mu)}{f_g(\varepsilon+\mu)} \mid \mu\right]\right] \\
&= \mathbb{E}_g\left[\sigma^2 \mathbb{E}\left[\frac{d}{d\varepsilon}\frac{f_g'(\varepsilon+\mu)}{f_g(\varepsilon+\mu)} \mid \mu\right]\right] \\
&= \mathbb{E}_g\left[\sigma^2 \left(\frac{f_g''(Z)}{f_g(Z)} - \frac{f_g'(Z)^2}{f_g(Z)^2}\right)\right] \\
&= -\sigma^2 I(f_g)
\end{aligned}
$$

The last step that remains to be shown is that $\mathbb{E}_g\left[\frac{f_g''(Z)}{f_g(Z)}\right] = 0$. But this is very similar to a standard Fisher information calculation, in which we interchange integration and differentiation to get that (here $\mu \sim g$):

$$
\begin{aligned}
\mathbb{E}_g\left[\frac{f_g''(Z)}{f_g(Z)}\right] &= \int f_g''(z)dz = \frac{1}{\sqrt{2\pi\sigma^2}}\int \frac{d^2}{dz^2}\mathbb{E}_g\left[\phi((z-\mu)/\sigma)\right]dz = \frac{1}{\sqrt{2\pi\sigma^2}}\int \mathbb{E}_g\left[\frac{d^2}{dz^2}\phi((z-\mu)/\sigma)\right]dz \\
&= \frac{1}{\sqrt{2\pi\sigma^2}}\mathbb{E}_g\left[\int \frac{d^2}{dz^2}\phi((z-\mu)/\sigma)dz\right] = 0
\end{aligned}
$$

$\square$

## A.5 Local Fano's Lemma

In this section we provide a Lemma to lower bound the expression $\inf_{\hat{V}_n} \mathbb{P}\left[\hat{V}_n \neq V_n\right]$ which appears in Lemma 1. Below, we denote by $\mathbb{P}^X \otimes \mathcal{N}\left(m_v(\cdot),\ \sigma^2 + A\right)$ the joint distribution of $(X, Z)$ when $X \sim \mathbb{P}^X$ and $Z \mid X \sim \mathcal{N}\left(m_v(X),\ \sigma^2 + A\right)$.

**Lemma 14** (Local Fano). *Assume there exists $\kappa > 0$ such that for all $v, v' \in \mathcal{V}_n$:*

$$
D_{KL}\left(\mathbb{P}^X \otimes \mathcal{N}\left(m_v(\cdot),\ \sigma^2 + A\right) \ \|\ \mathbb{P}^X \otimes \mathcal{N}\left(m_{v'}(\cdot),\ \sigma^2 + A\right)\right) \leq \kappa^2
$$

*If also:*

$$
\log(|\mathcal{V}_n|) \geq 2(n\kappa^2 + \log(2))
$$

*Then:*

$$
\inf_{\hat{V}_n} \mathbb{P}\left[\hat{V}_n \neq V_n\right] \geq \frac{1}{2}
$$

*Proof.* Let $V_n$ uniformly distributed on $\mathcal{V}_n$ and $\hat{V}_n$ any estimator of $V_n$. Then by Fano's inequality (Corollary 7.9 in Duchi [2019]):

$$
\mathbb{P}\left[\hat{V}_n \neq V_n\right] \geq 1 - \frac{I(V_n; (X_i, Z_i)_{1 \leq i \leq n}) + \log(2)}{\log(|\mathcal{V}_n)|}
$$

Here $I(V_n; (X_i, Z_i)_{1 \leq i \leq n})$ is the mutual information between $V_n$ and $(X_i, Z_i)_{1 \leq i \leq n}$.

Next fix $v, v' \in \mathcal{V}_n$ and let $P_v, P_{v'}$ the induced distributions of $(X_1, Z_1)$ induced by $m_v$, resp. $m_{v'}$ in model (1), then by (7.4.5) in Duchi [2019]:

$$
I(V_n; (X_i, Z_i)_{1 \leq i \leq n}) \leq \frac{1}{|\mathcal{V}|^2}\sum_{v, v' \in \mathcal{V}_n} D_{\mathrm{KL}}(P_v^n \| P_{v'}^n) \leq n \max_{v, v' \in \mathcal{V}_n} D_{\mathrm{KL}}(P_v \| P_{v'}) \leq n\kappa^2
$$

The result follows. $\square$

## A.6 Fay Herriot results

*Proof.* For the upper bound, we will use Theorem 2, where our regression estimator is just the ordinary least squares fit, i.e. $\hat{m}(x) = x^\top \hat{\beta}$ with $\hat{\beta} = (X^\top X)^{-1} X^\top Z_{1:n}$. By $X$ we mean the usual design matrix in which the vectors $X_1, \ldots, X_n$ are stacked as rows into a matrix.

We start by decomposing the error:

$$
\begin{aligned}
\mathbb{E}\left[(\hat{m}(X_{n+1}) - m(X_{n+1}))^2\right] &= \mathbb{E}\left[\left(X_{n+1}^\top \hat{\beta} - X_{n+1}^\top \beta\right)^2\right] \\
&= \mathbb{E}\left[\operatorname{tr}\left((\hat{\beta} - \beta)^\top X_{n+1} X_{n+1}^\top (\hat{\beta} - \beta)\right)\right] \\
&= \mathbb{E}\left[\operatorname{tr}\left((\hat{\beta} - \beta)(\hat{\beta} - \beta)^\top X_{n+1} X_{n+1}^\top\right)\right] \\
&= \operatorname{tr}\left(\mathbb{E}\left[(\hat{\beta} - \beta)(\hat{\beta} - \beta)^\top\right] \Sigma\right)
\end{aligned}
$$

Hence recalling that $\mathbb{E}\left[\hat{\beta}\right] = \beta$, we only need to study the covariance of $\hat{\beta}$.

$$
\begin{aligned}
\operatorname{Cov}\left[\hat{\beta}\right] &= \mathbb{E}\left[\operatorname{Cov}\left[\hat{\beta} \mid X_{1:n}\right]\right] + \operatorname{Cov}\left[\mathbb{E}\left[\hat{\beta} \mid X_{1:n}\right]\right] \\
&= (\sigma^2 + A)\mathbb{E}\left[(X^\top X)^{-1}\right] + 0 \\
&= (\sigma^2 + A)\Sigma^{-1}\frac{1}{n - d - 1}
\end{aligned}
$$

The last equality holds because $X^\top X$ follows a Wishart distribution. See Theorem 2 in Rosset and Tibshirani [2018] and references therein for similar results. In total we get:

$$
\mathbb{E}\left[(\hat{m}(X_{n+1}) - m(X_{n+1}))^2\right] = \operatorname{tr}\left((\sigma^2 + A)\Sigma^{-1}\frac{1}{n - d - 1}\Sigma\right) = \frac{d(\sigma^2 + A)}{n - p - 1}
$$

For the lower bound, we will apply Lemma 1. First we let $\mathcal{V}_n$ be an $1/2$ packing of the Euclidean ($\ell_2$) unit ball which has cardinality at least $2^d$ (Lemma 7.6. in Duchi [2019])

Then, for $v \in V_n$ we define $\theta_v = \varepsilon v$ (we will specify $\varepsilon$ later). Then we let $\beta_v = \Sigma^{-1/2}\theta_v$ and note that for two distinct $v, v'$:

$$
\begin{aligned}
\mathbb{E}\left[\left(X_{n+1}^\top \beta_v - X_{n+1}^\top \beta_{v'}\right)^2\right] &= \operatorname{tr}\left(\mathbb{E}\left[(\beta_v - \beta_{v'})(\beta_v - \beta_{v'})^\top\right] \Sigma\right) \\
&= \operatorname{tr}\left(\mathbb{E}\left[\Sigma^{-1/2}(\theta_v - \theta_{v'})(\theta_v - \theta_{v'})^\top \Sigma^{-1/2}\right] \Sigma\right) \\
&= \mathbb{E}\left[\|\theta_v - \theta_{v'}\|_2^2\right] \\
&\geq \frac{\varepsilon^2}{4}
\end{aligned}
$$

In the last step we used the packing property of the set $\mathcal{V}_n$ we defined.

On the other hand:

$$
\begin{aligned}
&D_{\mathrm{KL}}\left(\mathcal{N}(0, \Sigma) \otimes \mathcal{N}(\langle \cdot, \beta_v\rangle, \sigma^2 + A) \;\|\; \mathcal{N}(0, \Sigma) \otimes \mathcal{N}(\langle \cdot, \beta_{v'}\rangle, \sigma^2 + A)\right) \\
=&\mathbb{E}\left[D_{\mathrm{KL}}\left(\mathcal{N}(X_{n+1}^\top \beta_v, \sigma^2 + A) \;\|\; \mathcal{N}(X_{n+1}^\top \beta_{v'}, \sigma^2 + A) \mid X_{n+1}\right)\right] \\
=&\mathbb{E}\left[\frac{1}{2(A + \sigma^2)}\left(X_{n+1}^\top \beta_v - X_{n+1}^\top \beta_{v'}\right)^2\right] \\
=&\frac{1}{2(A + \sigma^2)}\mathbb{E}\left[\|\theta_v - \theta_{v'}\|_2^2\right] \\
\leq&\frac{2\varepsilon^2}{A + \sigma^2}
\end{aligned}
$$

To apply Lemma 14 we need the following to hold for a constant $C$:

$$
\log(2^d) \geq C\frac{n\varepsilon^2}{A + \sigma^2}
$$

So we may pick $\varepsilon^2 = c\frac{d(A+\sigma^2)}{n}$ for a constant $c$. Since $\varepsilon \to 0$ as $n \to \infty$, we may apply Lemma 1 for large enough $n$ with separation say $\varepsilon^2/10$, by which we can conclude. $\qquad\square$

### A.7 Lipschitz results

*Proof.* The upper bound follows from Theorem 2, where the regressor $\hat{m}_n$ is the $k$-nearest neighbor regression predictor (KNN) with optimally tuned number of neighbors, see Theorem 6.2 and Problem 6.7 in Györfi et al. [2006].

For the lower bound, we will apply Lemma 1. To this end, we start by constructing $\mathcal{V}_n$ as in the proof of Theorem 3.2. in Györfi et al. [2006]: We define $M_n \in \mathbb{N}$ and partition $[0,1]^d$ (we will pick $M_n$ later) into $M_n^d$ cubes $A_{n,j}$ of side length $1/M_n$ and with centers $a_{n,j}$. Next we take any function $\bar{m} : \mathbb{R}^d \to \mathbb{R}$ which is 1-Lipschitz, vanishes outside $[-\frac{1}{2}, \frac{1}{2}]^d$ and $C_I := \int \bar{m}^2(x)dx > 0$. We also define $\bar{m}_L(\cdot) = L \cdot \bar{m}(\cdot)$. Finally, for $j = 1, \ldots, M_n^d$ we define:

$$\bar{m}_{L,n,j}(x) = \frac{1}{M_n}\bar{m}_L(M_n(x - a_{n,j}))$$

Then we let $\mathcal{V}_n \subset \{\pm 1\}^{M_n^d}$ with $|\mathcal{V}_n| \geq \exp(M_n^d/8)$ and so that for all $v, v' \in \mathcal{V}_n$:

$$\sum_{j=1}^{M_n^d} \mathbf{1}\left(v_j \neq v_j'\right) \geq \frac{M_n^d}{4}$$

Such a set exists by the Gilbert-Varshamov bound (Lemma 7.5 in Duchi [2019]). With $\mathcal{V}_n$ in hand, we define for $v \in \mathcal{V}_n$:

$$m_v(x) = \sum_{j=1}^{M_n^d} v_j \bar{m}_{L,n,j}(x)$$

We argue that $m_v(x)$ indeed is $L$-Lipschitz: All $\bar{m}_{L,n,j}$ are $L$-Lipschitz, since so is $\bar{m}_L$ and furthermore observe that all $\bar{m}_{L,n,j}$, $j = 1, \ldots, M_n^d$ have disjoint support.

Next, take $v \neq v' \in \mathcal{V}_n$. Then, since the $\bar{m}_{L,n,j}$ have disjoint support:

$$
\begin{aligned}
\int (m_v(x) - m_{v'}(x))^2 \, d\mathbb{P}^X(x) &= \sum_{j=1}^{M_n^d} (v_j - v_j')^2 \int \bar{m}_{L,n,j}^2(x) d\mathbb{P}^X(x) \\
&\geq \sum_{j=1}^{M_n^d} (v_j - v_j')^2 \eta \int \bar{m}_{L,n,j}^2 dx \\
&= \sum_{j=1}^{M_n^d} (v_j - v_j')^2 \frac{\eta L^2}{M_n^{2+d}} C_I \\
&= \frac{4\eta L^2}{M_n^{2+d}} C_I \sum_{j=1}^{M_n^d} \mathbf{1}\left(v_j \neq v_j'\right) \\
&\geq \frac{4\eta L^2}{M_n^{2+d}} C_I \frac{M_n^d}{4} = \eta C_I \frac{L^2}{M_n^2}
\end{aligned}
$$

On the other hand, let us bound the KL divergence between the distributions induced by $m_v, m_{v'}$:

$$
\begin{aligned}
&D_{\mathrm{KL}}\left(\mathbb{P}^X \otimes \mathcal{N}\left(m_v(\cdot),\ \sigma^2 + A\right)\ \|\ \mathbb{P}^X \otimes \mathcal{N}\left(m_{v'}(\cdot),\ \sigma^2 + A\right)\right) \\
&= \mathbb{E}\left[D_{\mathrm{KL}}\left(\mathcal{N}\left(m_v(X_{n+1}),\ \sigma^2 + A\right)\ \|\ \mathcal{N}\left(m_{v'}(X_{n+1}),\ \sigma^2 + A\right) \mid X_{n+1}\right)\right] \\
&= \int \frac{1}{2(\sigma^2 + A)}\left(m_v(x) - m_{v'}(x)\right)^2 d\mathbb{P}^X(x) \\
&\leq \frac{1}{2\eta(\sigma^2 + A)}\int \left(m_v(x) - m_{v'}(x)\right)^2 dx \\
&\leq \frac{1}{2\eta(\sigma^2 + A)}\frac{4L^2}{M_n^{2+d}} C_I \sum_{j=1}^{M_n^d} \mathbf{1}\left(v_j \neq v_j'\right) \\
&\leq \frac{2C_I}{\eta(\sigma^2 + A)}\frac{L^2}{M_n^2}
\end{aligned}
$$

Next, we will lower bound $\inf_{\hat{V}_n} \mathbb{P}\left[\hat{V}_n \neq V_n\right]$ by Lemma 14. To get the condition, we need that for some $C > 0$:

$$
M_n^d \geq C \frac{L^2 n}{(\sigma^2 + A)M_n^2} \Leftrightarrow M_n \geq C\left(\frac{L^2 n}{\sigma^2 + A}\right)^{\frac{1}{2+d}}
$$

Hence for some $C$, we set $M_n = \lceil C\left(\frac{L^2 n}{\sigma^2 + A}\right)^{\frac{1}{2+d}}\rceil$. Then the separation between two hypotheses $m_v, m_{v'}$ is equal to (for another constant $C'$):

$$
\int \left(m_v(x) - m_{v'}(x)\right)^2 d\mathbb{P}^X(x) \geq \eta C_I \frac{L^2}{M_n^2} \geq C'\left(\frac{L^d(\sigma^2 + A)}{n}\right)^{\frac{2}{2+d}}
$$

We conclude by Lemma 1 upon noting that $M_n \to \infty$ and hence $\sup_{v \in \mathcal{V}_n} \sup_x |m_v(x)| \to 0$ as $n \to \infty$.

$\square$

# B   Results for sample-split EB in Section 3

in Section 3 we made the following point: Even if we knew the true $A$, it would not be the optimal $A$ to plug into (7). We formalize this in the following proposition:

**Proposition 15.** *Consider model* (1)*. Fix any (deterministic) function* $\tilde{m} : \mathcal{X} \to \mathbb{R}$ *and define:*

$$
A_{\tilde{m}} := \mathbb{E}_{m,A}\left[\left(\tilde{m}(X_{n+1}) - Z_{n+1}\right)^2\right] - \sigma^2 \tag{20}
$$

*Then:*

$$
\mathbb{E}_{m,A}\left[\left(t_{\tilde{m}, A_{\tilde{m}}}^*(X_{n+1}, Z_{n+1}) - \mu_{n+1}\right)^2\right] = \inf_{\tilde{A} \geq 0} \mathbb{E}_{m,A}\left[\left(t_{\tilde{m}, \tilde{A}}^*(X_{n+1}, Z_{n+1}) - \mu_{n+1}\right)^2\right]
$$

*The above expressions are equal to:* $\frac{\sigma^2 A_{\tilde{m}}}{\sigma^2 + A_{\tilde{m}}}$*. Furthermore, a direct consequence is that:*

$$
\mathbb{E}_{m,A}\left[\left(t_{\tilde{m}, A_{\tilde{m}}}^*(X_{n+1}, Z_{n+1}) - \mu_{n+1}\right)^2\right] \leq \mathbb{E}_{m,A}\left[\left(t_{\tilde{m}, A}^*(X_{n+1}, Z_{n+1}) - \mu_{n+1}\right)^2\right]
$$

*Proof.* Let us consider the following class of shrinkage rules, where $\lambda \in [0, 1]$:

$$
t_\lambda(x, z) = \lambda \tilde{m}(x) + (1 - \lambda)z = \lambda\left(\tilde{m}(x) - z\right) + z
$$

Then our goal will be to minimize the following function over $\lambda \in [0, 1]$:

$$
J(\lambda) = \mathbb{E}_{m,A}\left[\left(t_\lambda(X_{n+1}, Z_{n+1}) - \mu_{n+1}\right)^2\right] \tag{21}
$$

To this end:

$$J(\lambda) = \mathbb{E}_{m,A}\left[\{t_\lambda(X_{n+1}, Z_{n+1}) - \mu_{n+1}\}^2\right]$$

$$= \mathbb{E}_{m,A}\left[\{\lambda\left(\tilde{m}(X_{n+1}) - Z_{n+1}\right) + Z_{n+1} - \mu_{n+1}\}^2\right]$$

$$= \lambda^2 \mathbb{E}_{m,A}\left[(\tilde{m}(X_{n+1}) - Z_{n+1})^2\right] + 2\lambda\mathbb{E}_{m,A}\left[(\tilde{m}(X_{n+1}) - Z_{n+1})\left(Z_{n+1} - \mu_{n+1}\right)\right] + \sigma^2$$

$$= \lambda^2 \mathbb{E}_{m,A}\left[(\tilde{m}(X_{n+1}) - Z_{n+1})^2\right] - 2\lambda\sigma^2 + \sigma^2$$

The last step follows from the two following intermediate results:

$$\mathbb{E}_{m,A}\left[\tilde{m}(X_{n+1})\left(Z_{n+1} - \mu_{n+1}\right)\right] = \mathbb{E}_{m,A}\left[\tilde{m}(X_{n+1})\mathbb{E}_{m,A}\left[Z_{n+1} - \mu_{n+1} \mid X_{n+1}\right]\right] = 0$$

$$\mathbb{E}_{m,A}\left[Z_{n+1}\left(Z_{n+1} - \mu_{n+1}\right)\right] = \mathbb{E}_{m,A}\left[\mathrm{Var}_{m,A}\left[Z_{n+1} \mid \mu_{n+1}\right]\right] = \sigma^2$$

We may now directly minimizer over $A$ to see that the optimal $\lambda$ is given by:

$$\lambda^*(\tilde{m}) = \frac{\sigma^2}{\mathbb{E}_{m,A}\left[(\tilde{m}(X_{n+1}) - Z_{n+1})^2\right]}$$

The form of $A_{\tilde{m}}$ then directly follows by noting the one-to-one correspondence $\lambda \leftrightarrow \frac{\sigma^2}{A_{\tilde{m}}+\sigma^2}$. $\qquad\square$

We will now prove that for deterministic $\tilde{m}$, as in Proposition 15, parametric rates are possible in the estimation of $A_{\tilde{m}}$, which translate into $O(1/n)$ decay of the regret.

**Proposition 16.** *Consider $n$ i.i.d. observations $(X_i, Z_i)$ from model (1) with $A, \sigma > 0$. Fix any (deterministic) function $\tilde{m} : \mathcal{X} \to \mathbb{R}$ with $\mathbb{E}\left[\tilde{m}(X_{n+1})^4\right] \leq M$ for some $M < \infty$ (here $X_{n+1} \sim \mathbb{P}^X$). Let:*

$$\hat{A}_n = \left(\frac{1}{n}\sum_{k=1}^{n}(\tilde{m}(X_k) - Z_k)^2 - \sigma^2\right)_+$$

*Then $\hat{t}_n = t^*_{\tilde{m}, \hat{A}_n}$ satisfies:*

$$\mathbb{E}_{m,A}\left[L\left(\hat{t}_n; m, A\right)\right] \leq \mathbb{E}_{m,A}\left[L(t^*_{\tilde{m}, A_{\tilde{m}}}; m, A)\right] + O(1/n)$$

*Thus also:*

$$\mathbb{E}_{m,A}\left[L\left(\hat{t}_n; m, A\right)\right] \leq \mathbb{E}_{m,A}\left[L(t^*_{\tilde{m}, A}; m, A)\right] + O(1/n)$$

*Proof.* We consider again the $J(\lambda)$ from (21) and recall that $J(\lambda^*(\tilde{m})) = \min_{\lambda \geq 0} J(\lambda)$. We note that $J(\lambda)$ is a convex quadratic in $\lambda$ with:

$$J'(\lambda) = 2\lambda\mathbb{E}_{m,A}\left[(\tilde{m}(X_{n+1}) - Z_{n+1})^2\right] - 2\sigma^2, \quad J''(\lambda) = 2\mathbb{E}_{m,A}\left[(\tilde{m}(X_{n+1}) - Z_{n+1})^2\right]$$

Thus, since $J'(\lambda^*(\tilde{m})) = 0$, we get for any $\lambda$:

$$J(\lambda) = J(\lambda^*(\tilde{m})) + \mathbb{E}_{m,A}\left[(\tilde{m}(X_{n+1}) - Z_{n+1})^2\right](\lambda - \lambda^*(\tilde{m}))^2$$

This means that:

$$L(t_\lambda; m, A) = L(t_{\lambda^*(\tilde{m})}; m, A) + \mathbb{E}_{m,A}\left[(\tilde{m}(X_{n+1}) - Z_{n+1})^2\right](\lambda - \lambda^*(\tilde{m}))^2$$

Hence to conclude we will need to bound $\mathbb{E}_{m,A}\left[\left(\hat{\lambda}_n - \lambda^*(\tilde{m})\right)^2\right]$, where:

$$\hat{\lambda}_n = \frac{\sigma^2}{\sigma^2 \vee \left(\frac{1}{n}\sum_{k=1}^{n}(\tilde{m}(X_k) - Z_k)^2\right)}$$

Using the fact that both $\sigma^2 \vee \left( \frac{1}{n} \sum_{k=1}^{n} (\tilde{m}(X_k) - Z_k)^2 \right)$ and $\mathbb{E}_{m,A} \left[ (\tilde{m}(X_k) - Z_k)^2 \right]$ are $\geq \sigma^2$ and Taylor's theorem applied to $u \mapsto 1/u$, we get:

$$
\mathbb{E}_{m,A} \left[ \left( \hat{\lambda} - \lambda^*(\tilde{m}) \right)^2 \right] = \sigma^4 \mathbb{E}_{m,A} \left[ \left( \frac{1}{\sigma^2 \vee \left( \frac{1}{n} \sum_{k=1}^{n} (\tilde{m}(X_k) - Z_k)^2 \right)} - \frac{1}{\mathbb{E}_{m,A} \left[ (\tilde{m}(X_k) - Z_k)^2 \right]} \right)^2 \right]
$$

$$
\leq \mathbb{E}_{m,A} \left[ \left( \sigma^2 \vee \left( \frac{1}{n} \sum_{k=1}^{n} (\tilde{m}(X_k) - Z_k)^2 \right) - \mathbb{E}_{m,A} \left[ (\tilde{m}(X_k) - Z_k)^2 \right] \right)^2 \right]
$$

$$
\leq \mathbb{E}_{m,A} \left[ \left( \frac{1}{n} \sum_{k=1}^{n} (\tilde{m}(X_k) - Z_k)^2 - \mathbb{E}_{m,A} \left[ (\tilde{m}(X_k) - Z_k)^2 \right] \right)^2 \right]
$$

$$
= \text{Var}_{m,A} \left[ \frac{1}{n} \sum_{k=1}^{n} (\tilde{m}(X_k) - Z_k)^2 \right]
$$

$$
= \frac{1}{n} \text{Var}_{m,A} \left[ (\tilde{m}(X_k) - Z_k)^2 \right]
$$

This is $O(1/n)$ as long as $\text{Var}_{m,A} \left[ (\tilde{m}(X_k) - Z_k)^2 \right]$ is upper bounded, which is the case under the given assumptions. The last statement follows from Proposition 15.

$\square$

We are now in a position to prove Theorem 5

*Theorem 5.* We apply Proposition 16 for the data in fold $I_2$ conditionally on the first fold, i.e. conditionally on $Z_{I_1}, \mu_{I_1}, X_{I_1}$.

$\square$

## C   Results under misspecification

### C.1   Proof of Theorem 6 (James-Stein property)

Before proceeding with the proof, let us introduce the following lemma:

**Lemma 17.** *Fix $\nu \in \mathbb{N}$, a fixed vector $\xi = (\xi_1, \ldots, \xi_\nu)$, a mean vector $\theta = (\theta_1, \ldots, \theta_\nu)$ and independent $Y_1, \ldots, Y_\nu$ distributed as $Y_i \sim \mathcal{N}\left( \theta_i, \sigma^2 \right)$. Then consider the following positive-part James-Stein type estimator, parametrized by $a > 0$:*

$$
\hat{\theta}_a = \xi + \left( 1 - \frac{a\sigma^2}{\|Y - \xi\|_2^2} \right)_+ (Y - \xi) \tag{22}
$$

*This estimator has risk:*

$$
\mathbb{E} \left[ \left\| \hat{\theta}_a - \theta \right\|_2^2 \right] \leq \nu\sigma^2 - a\sigma^2 \left[ 2(\nu - 2) - a \right] \mathbb{E} \left[ \frac{\sigma^2}{\|Y - \xi\|_2^2} \right] \tag{23}
$$

*In particular, if $\nu \geq 5$ (resp. $\nu \geq 3$), $\hat{\theta}_\nu$ (resp. $\hat{\theta}_{\nu-2}$) has squared error risk $< \nu\sigma^2$.*

*Proof.* Estimator (22) where we do not take the positive part of $\left( 1 - \frac{a\sigma^2}{\|Y - \xi\|_2^2} \right)_+$ has risk precisely equal to the RHS in (23). This is well known, see for example Lemma 1 in [Green and Strawderman, 1991] and references therein. The positive part estimator then has smaller risk, as also follows from well known results on James-Stein estimation, see e.g. [Baranchik, 1964]. Finally when $a = \nu \geq 5$, $a\sigma^2 \left[ 2(\nu - 2) - a \right] = \sigma^2 \left[ \nu - 4 \right] > 0$.

$\square$

We are ready to prove Theorem 6:

*Proof.* Let $\widetilde{\mathbb{P}}_{I_1}[\,\cdot\,] = \mathbb{P}[\,\cdot\mid Z_{I_1}, \mu_{1:n}, X_{1:n}]$. Then w.r.t. $\widetilde{\mathbb{P}}_{I_1}[\cdot]$, it holds that $(Z_i)_{i\in I_2}$ are independent and $Z_i \sim \mathcal{N}\left(\mu_i, \sigma^2\right)$ for $i \in I_2$. Furthermore $\hat{m}_{I_1}(X_{I_2}) = (\hat{m}_{I_1}(X_i))_{i\in I_2}$ is deterministic w.r.t. $\widetilde{\mathbb{P}}_{I_1}[\cdot]$ and also recall that:

$$\hat{\mu}_{I_2}^{\text{EBCF}} = \frac{\sigma^2}{\hat{A}_{I_2} + \sigma^2}\hat{m}_{I_1}(X_{I_2}) + \frac{\hat{A}_{I_2}}{\hat{A}_{I_2} + \sigma^2}Z_{I_2}$$

$$= \hat{m}_{I_1}(X_{I_2}) + \left(1 - \frac{\sigma^2}{\hat{A}_{I_2} + \sigma^2}\right)(Z_{I_2} - \hat{m}_{I_1}(X_{I_2}))$$

Also from (8) it holds that:

$$\hat{A}_{I_2} = \left(\frac{1}{|I_2|}\sum_{i\in I_2}(\hat{m}_{I_1}(X_i) - Z_i)^2 - \sigma^2\right)_+$$

Thus $\hat{\mu}_{I_2}$ takes precisely the form from (22) with $a = |I_2|$ and thus applying Lemma 17 (w.r.t. $\widetilde{\mathbb{P}}_{I_1}[\,\cdot\,]$, also by assumption $|I_2| \geq 5$), we get:

$$\sum_{i\in I_2}\widetilde{\mathbb{E}}_{I_1}\left[\left(\mu_i - \hat{\mu}_i^{\text{EBCF}}\right)^2\right] < \sum_{i\in I_2}\widetilde{\mathbb{E}}_{I_1}\left[(\mu_i - Z_i)^2\right] = |I_2|\,\sigma^2$$

Integrate w.r.t. $Z_{I_1}$, to get:

$$\sum_{i\in I_2}\mathbb{E}\left[\left(\mu_i - \hat{\mu}_i^{\text{EBCF}}\right)^2\mid \mu_{1:n}, X_{1:n}\right] < |I_2|\,\sigma^2$$

Now apply the symmetric argument with the folds flipped to also get:

$$\sum_{i\in I_1}\mathbb{E}\left[\left(\mu_i - \hat{\mu}_i^{\text{EBCF}}\right)^2\mid \mu_{1:n}, X_{1:n}\right] < |I_1|\,\sigma^2$$

Add both inequalities and divide by $n$ to conclude. $\qquad\square$

## C.2   SURE results

Below we prove Theorem 7. Throughout the proof we deal with the more general case of unequal variances. In particular, we replace the assumption that $(X_i, Z_i)$ satisfy (9) by the following model (while keeping all other assumptions):

$$(X_i, \mu_i) \sim \mathbb{P}^{(X_i, \mu_i)}, \;\; Z_i \mid \mu_i, X_i \sim \left(\mu_i, \sigma_i^2\right), \;\; \text{i.e. } \mathbb{E}\left[Z_i \mid \mu_i, X_i\right] = \mu_i, \; \text{Var}\left[Z_i \mid \mu_i, X_i\right] = \sigma_i^2$$

*Theorem 7.* Our proof closely follows Xie et al. [2012]. Let $n_2 = |I_2|$. We also use the same notation as in the proof of Theorem 6, wherein $\widetilde{\mathbb{P}}_{I_1}[\,\cdot\,] = \mathbb{P}[\,\cdot\mid Z_{I_1}, \mu_{1:n}, X_{1:n}]$. For $i \in I_2$ we also write $\tilde{m}_i = \hat{m}_{I_1}(X_i)$. We rewrite the SURE expression as follows:

$$\text{SURE}_{I_2}(A) = \frac{1}{n_2}\sum_{i\in I_2}\left(\sigma_i^2 + \frac{\sigma_i^4}{(A + \sigma_i^2)^2}(Z_i - \tilde{m}_i)^2 - 2\frac{\sigma_i^4}{A + \sigma_i^2}\right) = \frac{1}{n_2}\sum_{i\in I_2}\left[\frac{\sigma_i^4}{(A + \sigma_i^2)^2}(Z_i - \tilde{m}_i)^2 + \frac{\sigma_i^2(A - \sigma_i^2)}{A + \sigma_i^2}\right]$$

We also define $\ell_{I_2}(A)$, the average loss in fold $I_2$ when we estimate $\mu_i$ by $t_{\hat{m}_{I_1}, A}^*(X_i, Z_i)$, i.e.:

$$\ell_{I_2}(A) := \frac{1}{n_2}\sum_{i\in I_2}\left(\mu_i - t_{\hat{m}_{I_1}, A}^*(X_i, Z_i)\right)^2$$

Next we collect the difference between the SURE risk estimate and the actual loss:

$$\text{SURE}_{I_2}(A) - \ell_{I_2}(A) = \frac{1}{n_2} \sum_{i \in I_2} \left[ \left( \frac{\sigma_i^4}{(A+\sigma_i^2)^2} (Z_i - \tilde{m}_i)^2 + \frac{\sigma_i^2(A-\sigma_i^2)}{A+\sigma_i^2} \right) - \left( \mu_i - \frac{A}{A+\sigma_i^2} Z_i - \frac{\sigma_i^2}{A+\sigma_i^2} \tilde{m}_i \right)^2 \right]$$

$$= \frac{1}{n_2} \sum_{i \in I_2} \left\{ \left[ (Z_i - \tilde{m}_i)^2 - \sigma_i^2 - (\mu_i - \tilde{m}_i)^2 \right] - \frac{2A}{A+\sigma_i^2} \left[ (Z_i - \tilde{m}_i)^2 - (Z_i - \tilde{m}_i)(\mu_i - \tilde{m}_i) - \sigma_i^2 \right] \right\}$$

$$= \text{I} + \text{II}$$

We consider each term independently. The first term does not depend on $A$, hence is easier to study.

$$\widetilde{\mathbb{E}}_{I_1} \left[ \left| \sum_{i \in I_2} \left[ (Z_i - \tilde{m}_i)^2 - \sigma_i^2 - (\mu_i - \tilde{m}_i)^2 \right] \right| \right]^2$$

$$\leq \widetilde{\mathbb{E}}_{I_1} \left[ \left( \sum_{i \in I_2} \left[ (Z_i - \tilde{m}_i)^2 - \sigma_i^2 - (\mu_i - \tilde{m}_i)^2 \right] \right)^2 \right]$$

$$= \sum_{i \in I_2} \widetilde{\text{Var}}_{I_1} \left[ (Z_i - \tilde{m}_i)^2 \right]$$

$$\leq 8 \sum_{i \in I_2} \left( \widetilde{\mathbb{E}}_{I_1} \left[ Z_i^4 \right] + \tilde{m}_i^4 \right) \leq 8 n_2 \left( \Gamma^4 + M^4 \right)$$

The second term depends on $A$ and we want a result that is uniform in $A$. Without loss of generality, we may assume that the indices in $I_2 = \{i_1, i_2, \dots\}$ are arranged such that $\sigma_{i_1}^2 \leq \sigma_{i_2}^2 \leq \dots$ (otherwise we may just rearrange). Then, as observed in Li [1986], Xie et al. [2012]:

$$\sup_{0 \leq A \leq \infty} \left| \sum_{i \in I_2} \frac{A}{A+\sigma_i^2} \left[ (Z_i - \tilde{m}_i)^2 - (Z_i - \tilde{m}_i)(\mu_i - \tilde{m}_i) - \sigma_i^2 \right] \right|$$

$$\leq \sup_{0 \leq c_n \leq \dots \leq c_1 \leq 1} \left| \sum_{i \in I_2} c_i \left[ (Z_i - \tilde{m}_i)^2 - (Z_i - \tilde{m}_i)(\mu_i - \tilde{m}_i) - \sigma_i^2 \right] \right|$$

$$= \max_{j=1,\dots,n_2} \left| \underbrace{\sum_{k=1}^{j} \left[ (Z_{i_k} - \tilde{m}_{i_k})^2 - (Z_{i_k} - \tilde{m}_{i_k})(\mu_{i_k} - \tilde{m}_{i_k}) - \sigma_{i_k}^2 \right]}_{=:M_j} \right|$$

Next notice that $M_j, j = 1, \dots, n_2$ is a martingale w.r.t. $\widetilde{\mathbb{P}}_{I_1}[\cdot]$, so by the $L^2$ maximal inequality, for a constant $C > 0$:

$$\widetilde{\mathbb{E}}_{I_1} \left[ \max_{j=1,\dots,n_2} M_j^2 \right] \leq 4 \widetilde{\mathbb{E}}_{I_1} \left[ M_{n_2}^2 \right] = 4 \sum_{i \in I_2} \widetilde{\text{Var}}_{I_1} \left[ (Z_i - \tilde{m}_i)^2 - (Z_i - \tilde{m}_i)(\mu_i - \tilde{m}_i) - \sigma_i^2 \right]$$

$$\leq C n_2 (\Gamma^4 + M^4)$$

The results together imply that for a constant $C' > 0$:

$$\widetilde{\mathbb{E}}_{I_1} \left[ \sup_{A \geq 0} |\text{SURE}_{I_2}(A) - \ell_{I_2}(A)| \right] \leq C' \sqrt{\Gamma^4 + M^4} \frac{1}{\sqrt{n_2}}$$

But by definition of $\hat{A}_{I_2}$, $\text{SURE}_{I_2}(\hat{A}_{I_2}) \leq \inf_{A \geq 0} \text{SURE}_{I_2}(A)$ and so for any $A \geq 0$:

$$\widetilde{\mathbb{E}}_{I_1} \left[ \ell_{I_2}(\hat{A}_{I_2}) \right] \leq \widetilde{\mathbb{E}}_{I_1} \left[ \ell_{I_2}(A) \right] + \widetilde{\mathbb{E}}_{I_1} \left[ \sup_{A \geq 0} |\text{SURE}_{I_2}(A) - \ell_{I_2}(A)| \right] \leq \widetilde{\mathbb{E}}_{I_1} \left[ \ell_{I_2}(A) \right] + C' \sqrt{\Gamma^4 + M^4} \frac{1}{\sqrt{n_2}}$$

This holds for any $A \geq 0$, hence it remains valid after we take the infimum over $A \geq 0$. $\quad\square$

### C.3 Proof of Corollary 8

*Proof.* By Theorem 7:

$$\frac{2}{n}\sum_{i\in I_2}\mathbb{E}\left[(\mu_i - \hat{\mu}_i^{\text{EBCF}})^2 \mid X_{1:n}, \mu_{1:n}, Z_{I_1}\right] \leq \inf_{A\geq 0}\left\{\frac{2}{n}\sum_{i\in I_2}\mathbb{E}\left[\left(\mu_i - t^*_{\hat{m}_{I_1},A}(X_i, Z_i)\right)^2 \mid X_{1:n}, \mu_{1:n}, Z_{I_1}\right]\right\} + O\left(\frac{1}{\sqrt{n}}\right)$$

Next integrate over $X_{1:n}, \mu_{1:n}, Z_{I_1}$ and pull the $\inf$ outside of the expectation and use the fact that $(X_i, Z_i, \mu_i)$ are i.i.d. to get for fresh $(X_{n+1}, Z_{n+1})$:

$$\frac{2}{n}\sum_{i\in I_2}\mathbb{E}\left[(\mu_i - \hat{\mu}_i^{\text{EBCF}})^2\right] \leq \inf_{A\geq 0}\left\{\mathbb{E}\left[\left(\mu_{n+1} - t^*_{\hat{m}_{I_1},A}(X_{n+1}, Z_{n+1})\right)^2\right]\right\} + O\left(\frac{1}{\sqrt{n}}\right)$$

Then, make the choice $A = \mathbb{E}\left[(\hat{m}_{I_1}(X_{n+1}) - \mu_{n+1})^2\right]$ to get:

$$\frac{2}{n}\sum_{i\in I_2}\mathbb{E}\left[(\mu_i - \hat{\mu}_i^{\text{EBCF}})^2\right] \leq \frac{\sigma^2\mathbb{E}\left[(\hat{m}_{I_1}(X_{n+1}) - \mu_{n+1})^2\right]}{\sigma^2 + \mathbb{E}\left[(\hat{m}_{I_1}(X_{n+1}) - \mu_{n+1})^2\right]} + O\left(\frac{1}{\sqrt{n}}\right)$$

Repeat the same argument with $I_1, I_2$ flipped, add the results and divide by 2 to conclude. $\square$