[Reviews · NeurIPS 2019]

Reviewer 1



I like the overall setup and the analysis from the statistical point of view. The theoretical results are sound, though I think obtained mostly with standard methods. What I am missing most is motivation and empirical validation which is crucial for NeurIPS submission in my opinion. First of all, I am not aware of serious practical interest in accurate computation of mean ratings for recommender systems. That might be important in some biological applications but seems not to be on agenda for recommender systems. Furthermore, the experimental comparison is 1) Very limited: one synthetic example and one real-world dataset 2) Standard deviations are not provided in all the experiments. It is thus not clear, whether the experimental benefits of the proposed method are statistically significant.

Reviewer 2



This paper shows a lot of theoretical work related to empirical Bayes estimates using covariates. It also proposes EBCF estimate by splitting the sample into two parts, one of which is used for mean estimation and the other of which is for variance estimations. Many theoretical proofs have been provided and robustness has been shown. Experiments on both synthetic and real data demonstrate better prediction accuracy of EBCF compared to other related methods. I am not familiar with Empirical Bayes theories, so I cannot evaluate the originality, the technical quality and the significance. My review is focused on the experiment results. Concretely, can the authors quantify to what degree EBCF dominates the methods compared by showing standard errors of MSE? In addition, what does EBCF stand for?

Reviewer 3



I feel the treatment in this paper is somewhat contradictory to its setup. It is assumed that the density of X is uniformly lower bounded. This reduces the problem to standard linear/nonparametric regression and all these rates are somewhat known. The fact that A is unknown is, I feel, not a big deal because this A is "global" in the sense that all movies should suffer if A is large and vice versa. What is really interesting, which I think the authors correctly set up, is when some movies have many more ratings compared to the other movies. In this case, we have an extreme form of heteroscedasticity (i.e., V(x_i) ~ sqrt{N} * V(x_j), for example) and things become interesting. The following form of estimation error might be expected (e.g., for Lipschitz smooth functions): err(x_i) ~= 1/root{n_i} + n^{-1/3}, where n_i is the number of times x_i is reviewed, and n is the total number of reviews in the movie data base. Such results would be more interesting. The bottomline is that pointwise error asymptotics needs to be understood, because there should be a difference on estiming ratings of well-seen and less-seen movies.

Reviewer 4



The Authors provide a simple but powerful approach to empirical bayesian inference under rather broad assumptions. The method is relevant in settings where both a) standard statistical estimators (such as the average) can be evaluated and b) covariates can be used to train machine learning models to estimate the same value. The paper tries to solve this problem in the setting where the standard estimator is not reliable enough (e.g. because sample size is too small) and the covariates only give weak information on the target variable. The problem setting considered is highly relevant in many real-world settings. Considering the practical relevance and theoretical interest in empirical bayes methods, it seems quite surprising that this approach has not been investigated earlier (only for special cases such as linear models). As such, the presented paper fills a very important gap by giving the proposed method (that apparently has already been used in practice, as noted in the related-works section) a clear theoretical basis. The focus of the paper lies on the theoretical analysis; the main result are minimax bounds that explicitly incorporate the ML model, as well as (potentially unknown) model variances. Additional theoretical contributions are robustness under misspecification of the model (leading to very general results), as well as an analysis of the practical implementation of the proposed model. The paper concludes with an empirical evaluation on both synthetic and real-world data. The paper is very well written; even without strong background in the topic it is easy to follow, results are being discussed, explained and put into context. As a minor drawback, a discussion of potential disadvantages of the proposed method could have been enlightening.

[Author Response · NeurIPS 2019]

**Referee 1:** We thank the referee for thoughtful feedback. We will further emphasize in the introduction that the considered problem of estimating means with contextual side information is ubiquitous in real-world settings. In high-throughput *biological studies* it is important to estimate effect sizes for genes based on extremely small samples (e.g. 3 patients with cancer vs. 3 healthy) and there exists a rich annotation and categorization of genes to be used as covariates. In the *surveying of subpopulations* (e.g. by the US Census Bureau) one can improve noisy estimates (e.g. on average income) for small communities by using geographical and economic covariates. In *baseball statistics*, our methods can be used to improve estimation of the batting average of each player, say to predict their performance in the second half of a season from the first, and covariate information could include salary, team and previous performance.

We chose the MovieLens dataset because it is familiar to the NeurIPS community and illustrates our main statistical considerations; we do not argue for the use of EBCF in Recommender systems. By the same token, however, typical Recommender systems are not good at predicting average ratings for sparsely watched items. Say a user comes across a movie on their own, then an improved rating estimate could help them choose whether to watch that movie or not. Furthermore, improved rating estimators could be useful to rank, say, new indie movies based on few initial ratings.

As suggested by the referee we have added confidence intervals (CI) for all empirical results (see plot below). For Fig.3 we report standard CIs of half-width $2 \times$ s.e. (standard error), for Fig.2 the error bars are inflated to $10 \times$ s.e., since otherwise they would not be visible at all.

|  | All | Sci-Fi & Horror |
|---|---|---|
| Unbiased | 0.098 ($\pm$ 0.004) | 0.098 ($\pm$ 0.032) |
| XGBoost | 0.145 ($\pm$ 0.004) | 0.183 ($\pm$ 0.030) |
| SURE | 0.061 ($\pm$ 0.002) | 0.064 ($\pm$ 0.018) |
| EBCF | **0.055** ($\pm$ 0.002) | **0.052** ($\pm$ 0.012) |

We have also analyzed an additional real world dataset, the Communities and Crimes unnormalized dataset from the UCI repository. Our task is to predict the nonviolent crime rate per 1k population for each community. We make the problem harder by using hypergeometric sampling to subsample the population of each community to $B \in \{200, 500\}$. The mean squared errors and 95% CIs are as follows for $B = 200$: Unbiased $224(\pm16)$, XGBoost $168(\pm20)$, SURE $184(\pm19)$ and EBCF $149(\pm20)$. For $B = 500$ they are: Unbiased $92(\pm8)$, XGBoost $122(\pm15)$, SURE $86(\pm8)$ and EBCF $80(\pm9)$. The revised manuscript will contain the details of this analysis and additional simulation results.

**Referee 2:** We thank the referee for the feedback. EBCF is the proposed method and stands for Empirical Bayes with Cross-Fitting; we will further clarify in the text. Furthermore, we have added standard errors (see reply to Referee 1).

**Referee 3:** We want to clarify that our results are not just "standard linear/nonparametric regression" and allowing $A > 0$ is crucial to our method. Only in the case $A = 0$ do our results collapse to standard regression. Let us note some differences (which we will further clarify in the manuscript): First, our objective is to estimate the $\mu_i$ with small mean squared error (MSE), not the regression function $m(\cdot)$. Second, for $A > 0$ the MSE for estimating $\mu_i$ is strictly $> 0$ and cannot go down to 0 even as $n \to \infty$. In standard regression treatments the error does go to 0. Instead, in our minimax analysis we consider the difference between the mean squared error and the Bayes risk (the regret): We then proceed to show that when the covariates and $m(\cdot)$ satisfy classical regression assumptions (e.g. bounded $X_i$ and Lipschitz $m(\cdot)$), then the regret in estimating $\mu_i$ is precisely characterized by the familiar minimax rates for nonparametric regression. This result is *not* a trivial consequence of existing results in nonparametrics; it is instead one of our contributions through Lemma 1 and Theorem 2.

On the other hand, we agree with the referee that e.g. bounded $X_i$ seems restrictive. This is why we then tackle the problem of robustness to misspecification in Section 4. Here our results are novel even under the pure regression setting (i.e. $A = 0$). Theorem 6 holds under no assumptions on $\hat{m}(\cdot)$ (it could be a deep neural net or a k-NN regressor or ...), nor on $m(\cdot)$ nor on the support of $X_i$. Prop. 7 also makes no assumptions on the support of $X_i$.

Finally, we thank the referee for an excellent suggestion on strengthening our results; we present a simplified sketch here: Consider the setup of the paper with $n$ observations $Z_i$ with variance $\sigma^2$, but now the $n + 1$-th observation has variance $\tau^2$ which may be $\neq \sigma^2$. Then in the Lipschitz case, extending Theorem 4, we can prove a regret that scales as $\text{MSE} - \frac{A\tau^2}{A+\tau^2} \sim \frac{\tau^4}{A+\tau^2}(\frac{A+\sigma^2}{n})^{2/3}$ (note that $A\tau^2/(A + \tau^2)$ is the Bayes risk). Letting $\tau^2 \sim 1/N$ and rearranging, similar to the referee's insight, we get $\text{MSE}^{1/2} \sim N^{-1/2} + N^{-1}n^{-1/3}$. The rapid decay of the 2nd term in $N$ theoretically and quantitatively verifies our empirical results (e.g. Figs 2b,c): The quality of the regression fit is important only for small $N$ where we benefit most from empirical Bayes shrinkage, while for large $N$ it does not matter.

**Referee 4:** We thank the referee for the kind words and a very detailed summary of our contributions. One shortcoming of the method is that the covariates $X_i$ can modulate the effect size distribution only through the map $X_i \mapsto \mathcal{N}(m(X_i), A)$. We have comprehensively studied this case in the paper, however in future work we hope to explore additional effect modifications, for example $X_i \mapsto \mathcal{N}(0, A(X_i))$ could be more relevant for differential gene expression studies in biology. Heavy-tailed priors and priors with a point mass at 0 are also of interest.

[Meta-Review · NeurIPS 2019]

This theory paper provides a number of novel results, including theoretical analysis of minimax bounds and an empirical analysis, for combinations of relatively simple statistical estimators and machine learning models of covariate information. The paper shows that these combinations improve on both the simple estimator alone and the machine learning model alone. The main concern raised by the reviewers is that the paper provides limited empirical validation. I disagree with this assessment, as the paper should be seen as a machine learning theory paper. As the proposed framework includes a number of advanced machine learning models, including XGBoost it should be very relevant for the NeurIPS community. Additionally, it provides a empirical validations of their proposed approach on a simulated data set and on real-world MovieLens ratings.